# Any-Order Flexible Length Masked Diffusion

**Jaeyeon Kim**[1,★]   **Lee Cheuk-Kit**[1,2,★]   **Carles Domingo-Enrich**[3]   **Yilun Du**[1,2]

**Sham Kakade**[1,2]   **Timothy Ngotiaoco**[1,2]   **Sitan Chen**[1,†]   **Michael S. Albergo**[1,2,4,†]

[1]Harvard University  [2]Kempner Institute  [3]Microsoft Research New England
[4]Institute for Artificial Intelligence and Fundamental Interactions, MIT

## Abstract

Masked diffusion models (MDMs) have recently emerged as a promising alternative to autoregressive models over discrete domains. MDMs generate sequences in an any-order, parallel fashion, enabling fast inference and strong performance on non-causal tasks. However, a crucial limitation is that they do not support token insertions and are thus limited to *fixed-length* generations. To this end, we introduce **Flex**ible **M**asked **D**iffusion **M**odels (FlexMDMs), a discrete diffusion paradigm that simultaneously can model sequences of flexible length while provably retaining MDMs' flexibility of any-order inference. Grounded in an extension of the stochastic interpolant framework, FlexMDMs generate sequences by inserting mask tokens and unmasking them. Empirically, we show that FlexMDMs match MDMs in perplexity while modeling length statistics with much higher fidelity. On a synthetic maze planning task, they achieve $\approx 60\%$ higher success rate than MDM baselines. Finally, we show pretrained MDMs can easily be *retrofitted* into FlexMDMs: on 16 H100s, it takes only three days to fine-tune LLaDA-8B into a FlexMDM, achieving superior performance on math (GSM8K, 58%→67%) and code infilling performance (52%→65%). We open-source our code at https://github.com/brianlck/FlexMDM.

## 1 Introduction

While diffusion models (Ho et al., 2020; Song et al., 2020; Sohl-Dickstein et al., 2015) are now the leading paradigm for generative modeling in continuous domains, recent work has begun to expand their scope to discrete spaces. The prevailing approach, Masked Diffusion Models (MDMs) (Shi et al., 2024; Sahoo et al., 2024; Gat et al., 2024), generates sentences in a non-left-to-right, any-order fashion. Compared to autoregressive models, this any-order generation ability yields substantially faster inference and strong downstream performance on non-casual tasks such as planning (Ye et al., 2024), code Nie et al. (2025); Ye et al. (2025), and reasoning (Nie et al., 2024).

Despite these successes, current MDMs cannot (1) model distributions supported on sequences of *variable length* and (2) insert new tokens during generation (Figure 1, left). Both capabilities are natural desiderata for generative models over discrete domains. We therefore ask: *Can we model variable-length data while preserving MDMs' any-order generation power?*

We answer in the affirmative by proposing the **Flexible Masked Diffusion Model** (FlexMDM). FlexMDMs start from an empty string and gradually insert mask tokens and then unmask them (Figure 1, right). Beyond learning the usual *unmasking* posterior–the distribution of a clean token at masked positions–we introduce an *insertion* expectation: the expected number of tokens to insert conditioned on the current sequence. Crucially, we show that FlexMDM is theoretically grounded (i.e., under perfect training, it samples from the true data distribution) and retains the any-order sampling property of MDMs, thereby directly addressing the question above. Empirically, we demonstrate that FlexMDM offers significant new upgrades to the MDM paradigm,

---

★equal contribution, randomized ordering; Lee Cheuk-Kit led the theoretical development of the method and engineering effort; Jaeyeon Kim led experiment development and paper presentation. † lead senior authors.

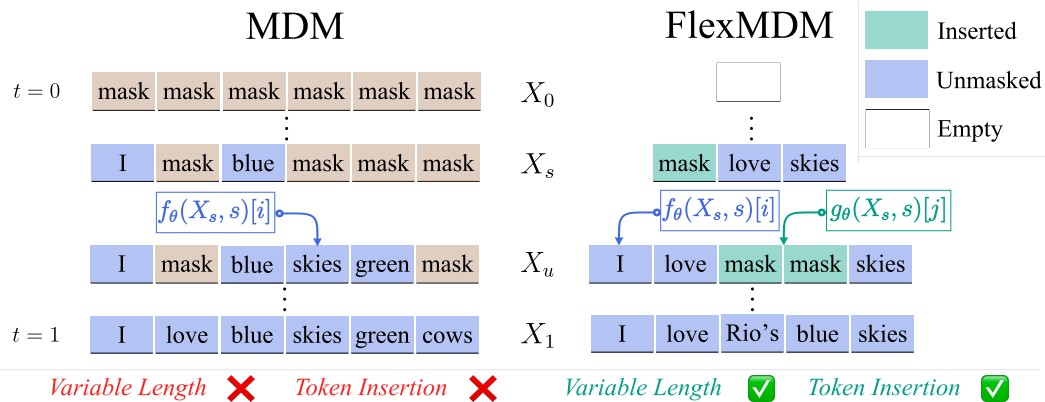

**Figure 1: Flexible Masked Diffusion Model** (FlexMDM) addresses MDMs' inability to handle variable-length sequences and token insertion while preserving any-order generation power. At each step, FlexMDM performs **insertion** and **unmasking** by predicting **the expected number of mask tokens to insert** ($g_\theta$) and the **posterior over clean tokens** ($f_\theta$), respectively.

- A FlexMDM pretrained on OpenWebText is able to model the length distribution with substantially higher fidelity while matching the perplexity of an MDM counterpart.
- On planning tasks, FlexMDM achieves markedly better results, beating the success rate of MDMs by nearly 60% on a natural synthetic baseline.
- MDMs can be retrofitted into FlexMDMs at 8B+ scale: We fine-tune LLaDA-8B (Nie et al., 2025), an open-source MDM, into a FlexMDM using only 16 H100s for three days. The model transfers cleanly from its MDM initialization and, with its newly acquired variable-length capability, attains notably better performance on GSM8K (58%→67%) and Code infilling (52%→65-%).

Theoretically, our construction relies on the machinery of continuous-time Markov chains (CTMCs) and in particular introduces the new notion of a *joint interpolant*, a novel extension of *stochastic interpolants* (Albergo & Vanden-Eijnden, 2022; Albergo et al., 2023b; Lipman et al., 2022). Recent work (Zheng et al., 2024; Ou et al., 2024) established an equivalence between MDMs and any-order language models–obviating the need for CTMCs. In contrast, we prove that FlexMDMs also possess the flexibility of any-order generation, yet the continuous-time perspective is absolutely essential for them to accurately model the length distribution. Accordingly, we re-derive the connections between MDMs and stochastic interpolants and use them to ground the design of FlexMDMs.

**Roadmap.** We begin in Section 2 with a broadly accessible review of CTMCs and the connection between MDMs and discrete flow matching. Building on this, Section 3 derives the FlexMDM training objective, and Section 4 introduces our inference procedures. Section 5 presents our experimental results.

**Concurrent work.** Concurrent works (Wu et al., 2025b; Havasi et al., 2025) attempt to tackle the same problem. Wu et al. (2025b) introduces an auxiliary expand token in training and heuristically replaces each expand token with two mask tokens at inference. Havasi et al. (2025), also based on the discrete flow matching framework, shares a similar theoretical grounding as our result. The main differences lie in our particular choice of interpolant that leads to the development of an any-order sampling algorithm. For clarity, we provide a detailed comparison in Appendix A.

## 2 PRELIMINARIES: CONTINUOUS-TIME MARKOV CHAINS AND MASKED DIFFUSIONS

In what follows, we provide an overview of continuous-time Markov chains (CTMCs), their role in defining discrete diffusion models, and link them to the MDM framework. As we mentioned in the introduction, this theme is essential to defining FlexMDMs in Section 3.

**Transport with continuous-time Markov Chains.** Given a target distribution $p_1$ over sequences with a finite vocabulary set (e.g., text), our aim is to learn to transport samples from a reference distribution $p_0$ through a continuum of distributions $\{p_t\}_{t\in[0,1]}$ such that $p_{t=1} = p_1$. This type of trans-

port can be realized by a continuous-time Markov chain, which is a stochastic process $\{X_t\}_{t\in[0,1]}$ with $X_0 \sim p_0$ governed by a time-dependent transition rate matrix $\{R_t(\cdot,\cdot)\}_{t\in[0,1]}$ satisfying

$$R_t(x,x) = -\sum_{y\neq x} R_t(x,y), \quad R_t(x,y) \geq 0, \ x \neq y. \tag{1}$$

Intuitively, the rate matrix determines the infinitesimal likelihood that $X_t$ transitions to any other state $y$ via

$$\mathbb{P}(X_{t+h} = y | X_t = x) = \mathbf{1}_{\{x=y\}} + hR_t(x,y) + o(h), \tag{2}$$

where we denote the conditional probability measure $\mathbb{P}(\cdot|X_t = x)$ of a new state given the present one. Here $o(h)$ is a remainder term that vanishes faster than $h$ as $h \to 0$. In generative modeling for these discrete distributions, our aims are to **(a)** specify a path of marginal distributions $\{p_t\}_{t\in[0,1]}$ connecting $p_0$ to $p_1$ and **(b)** learn the associated $R_t$ such that these marginals collectively satisfy the Kolmogorov forward equation:

$$\partial_t p_t(x) = \sum_y p_t(y)R_t(y,x) \qquad p_{t=0} = p_0. \tag{3}$$

This ensures that at time $t = 1$, the evolution specified by (2) results in a sample from the target distribution $p_1$. The rate matrices defined in this paper are sparse; therefore, we assume that the unspecified entries are 0 and the diagonal entries are defined through Equation (1).

## 2.1 MASKED DIFFUSION MODELS

We briefly review MDMs (Sahoo et al., 2024; Shi et al., 2024) and discrete flow matching with the masked construction (Gat et al., 2024), through the lens of stochastic interpolants. The target distribution $p_1$ assigns probability to length $L$ sequences. The base distribution $p_0$ employed by these models is the point mass distribution at the fully masked length-$L$ sequence $(\mathbf{m}, \ldots, \mathbf{m})$, where $\mathbf{m}$ is an auxiliary mask token.

To define the intermediate $\{p_t\}_{t\in[0,1]}$ that bridges the base and the target, we make use of a stochastic interpolant $\{x_t\}_{t\in[0,1]}$, a collection of random variables whose marginal distribution defines the continuum $\{p_t\}_{t\in[0,1]}$, i.e., $x_t \sim p_t$. Although the previous notion of stochastic interpolant (Albergo et al., 2023b) is defined in a continuous space, it naturally extends to a discrete space, and we defer a formal exposition to Appendix C.

**Design of distribution path.** The stochastic interpolant relies on a smooth and monotone unmasking schedule $\alpha_t \colon [0,1] \to [0,1]$ with boundary condition $(\alpha_0, \alpha_1) = (0,1)$ and time derivative denoted by $\dot{\alpha}_t$. To draw $x_t$, we first sample a clean sequence $x_1 \sim p_1$; then, independently for every coordinate $i$, we draw an unmasking time $T^i$ from density $\dot{\alpha}_t dt$ and set

$$x_t^i = \begin{cases} \mathbf{m} & t < T^i \\ x_1^i & t \geq T^i \end{cases}.$$

This process is illustrated in Figure 2. Hence, each clean token stays masked with probability $1-\alpha_t$ in a coordinate-independent fashion, defining $p_t(\cdot \mid x_1)$. We then write $p_t$ by marginalizing over $x_1 \sim p_1$.

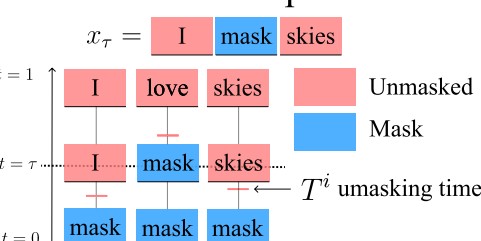

## MDM Interpolant

**Figure 2:** To draw a sample $x_t$, one can equivalently sample the clean sequence $x_1 \sim p_1$, draw unmasking times, and then accordingly unmask or mask each coordinate's token.

**MDM training.** We now derive the MDM rate matrix that induces a CTMC whose marginals coincide with $\{p_t\}_{t\in[0,1]}$ and how it is learned in practice. The central object is the unmasking posterior: the posterior on the clean token $x_1^i$ for masked index $i$ given $x_t = x$ and time step $t$, i.e., $\mathbb{P}(x_1^i = v | x_t = x)$. We model this posterior with a neural network $f_\theta(x,t) \in (\Delta(\Sigma))^n$, where $\Delta(\Sigma)$ denotes a simplex of probability distributions over the vocabulary $\Sigma$.

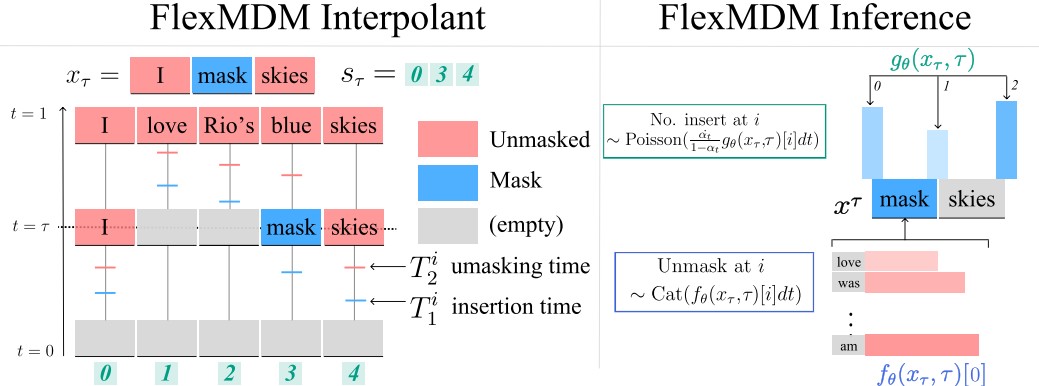

**Figure 3: Left (FlexMDM interpolant).** To draw a sample $x_t$, one can equivalently draw a sample $x_1 \sim p_1$, and for each token unmask, mask, or remove it according to the unmasking and insertion times $(T_1^i, T_2^i)$. An auxiliary interpolant $s_t$ gives closed-form expressions for the FlexMDM rate matrices. **Right (FlexMDM Inference).** Learned unmasking posterior and insertion expectation are later used at inference.

For every position where $x^i = \mathbf{m}$, the network aims to predict $f_\theta(x, t)[i, v] \approx \mathbb{P}(x_1^i = v | x_t = x)$, and is trained by minimizing the following variational loss:

$$\mathcal{L}_\theta = -\int_0^1 \mathbb{E}\left[ \frac{\dot{\alpha}_t}{1 - \alpha_t} \sum_{i:\ x_t^i = \mathbf{m}} \log f_\theta(x_t, t)[i, x_1^i] \right] dt. \tag{4}$$

Here, $\mathbb{E}$ denotes the expectation over $x_1 \sim p_1$ and $x_t \sim p_t(\cdot | x_1)$. The minimizer of this loss is the ground-truth unmasking posterior, which fully determines the MDM's rate matrix below. Precisely, for $t \in [0, 1]$, the rate matrix at time $t$ is given by: for a partially masked sequence $x \in (\Sigma \cup \{\mathbf{m}\})^L$,

$$R_t(x, x[x^i \leftarrow v]) = \frac{\dot{\alpha}_t}{1 - \alpha_t} \underbrace{\mathbb{P}(x_1^i = v | x_t = x)}_{\text{unmasking posterior}}, \quad v \in \Sigma, x^i = \mathbf{m}, \tag{5}$$

where $x[x^i \leftarrow v]$ denotes the sequence obtained from $x$ by replacing its $i$-th token with $v$. Therefore, once $f_\theta$ has learned the unmasking posterior, one can simulate the CTMC using the rate matrix in (5). The variational loss (4) quantifies the sampling guarantee of this *estimated* CTMC. Let $p_1^\theta$ be the terminal distribution of the estimated CTMC.Then, the loss function bounds the KL-divergence:

$$\mathcal{D}_{\text{KL}}(p_1 || p_1^\theta) \leq \mathcal{L}_\theta,$$

where $\mathcal{L}_\star$ is the global minimum of $\mathcal{L}$. When the loss is in its infimum, the KL divergence vanishes, resulting in the ground truth distribution.

**Connections to other MDM frameworks.** Connecting to prior works on MDM (Shi et al., 2024; Sahoo et al., 2024; Campbell et al., 2022), defining an interpolant is similar to defining a forward process for the case of diffusion models or a probability path in the case of flow matching. The modeled quantity is identical to the unmasking posterior across all frameworks. For inference, a common scheme is to proceed by: at each intermediate time step, (a) selecting a subset of indices to unmask and (b) sampling clean tokens from the learned posterior. In the infinitesimal limit, this procedure is equivalent to simulating the CTMC of (5). Meanwhile, subsequent work Kim et al. (2025); Nie et al. (2025) shows that MDMs also allow theoretically grounded *any-order inference*: tokens can be unmasked in an arbitrary order without necessarily following the CTMC at (5). We will revisit this aspect in Section 4 and show that our FlexMDM preserves this advantage.

## 3 Variable Length Masked Diffusions: Training

In this section, we introduce **Flexible Length Masked Diffusion Model** (FlexMDM): a discrete diffusion that models a distribution $p_1$ assigning probabilities to sequences of different lengths. Following the MDM's recipe, we aim to introduce a stochastic interpolant $x_t$ whose marginal distribution defines the path $\{p_t\}_{t \in [0,1]}$ and learn the corresponding CTMC. Everything hinges on selecting

an interpolant that is **(a)** easy to sample at $t = 0$ and **(b)** equipped with a closed-form rate matrix amenable to neural network training.

**Challenge.** Reusing the MDM interpolant is *inadequate*: at $t = 0$, the base distribution $p_0$ would consist of fully-masked sentences of *variable lengths*, which is impossible to sample since we do not know the length statistics of $p_1$ in advance. On the other hand, one can consider an interpolant constructed by masking and removing tokens from a clean sequence. However, this *complicates* the rate matrix characterization–token indices shift as insertions occur. To bridge this gap, we introduce the joint interpolant, an extension of the stochastic interpolant that augments the process with an auxiliary variable explicitly tracking token positions. This enlarged state space allows us to construct a broader class of rate matrices while preserving an easy-to-sample base distribution.

**Design of distribution path.** We now introduce our FlexMDM's joint interpolant that allows us to model the variable length $p_1$. This construction relies on *two* smooth, monotone schedules – an insertion schedule $\alpha \colon [0, 1] \to [0, 1]$ and an unmasking schedule $\beta \colon [0, 1] \to [0, 1]$, with the boundary conditions $(\alpha_0, \alpha_1) = (\beta_0, \beta_1) = (0, 1)$ and time derivatives denoted by $\dot{\alpha}_t, \dot{\beta}_t$.

To draw $x_t$, we first sample a clean sentence $x_1 \sim p_1$. Independently for each coordinate $i$, we draw an insertion time $T_1^i$ and an unmasking time $T_2^i$ with $T_1^i < T_2^i$ according to the density below. Accordingly, we either unmask, mask, or remove $x_1^i$ to obtain $x_t^i$:

$$T_1^i \sim \dot{\alpha}_t\, dt, \qquad T_2^i \sim \mathbf{1}_{\{t \geq T_1^i\}} \frac{\dot{\beta}_t}{1 - \beta_{T_1}}\, dt, \qquad x_t^i = \begin{cases} \text{(empty)}, & 0 < t < T_1^i \\ \mathbf{m}, & T_1^i \leq t < T_2^i \\ x_1^i, & T_2^i \leq t \leq 1 \end{cases} \tag{6}$$

Here, $\mathbf{1}$ denotes the indicator function. We obtain $x_t$ by concatenating the symbols $x_t^i$, and dropping $x_t^i = $ (empty). Consequently, the length of $x_t$ is equal to or less than that of $x_1$[1] (see Figure 3, left). As we mentioned above, we augment $x_t$ with an index-tracking variable $s_t$, forming the joint interpolant $(x_t, s_t)$. Let $\mathrm{len}(x_t)$ denote the length of $x_t$; then

$$s_t \colon = \{i \in \{0, \dots, \mathrm{len}(x_1) - 1\} \mid T_1^i \leq t\},$$

i.e., the set of indices whose clean tokens have *not* been deleted. Equivalently, the positions in $x_1$ referenced by $x_t$'s each index. By regarding $s_t$ as a list and ordering its elements in ascending order, we also have $x_t = (x_1^{s_t[0]}, \dots, x_1^{s_t[\mathrm{len}(s_t)-1]})$. We revisit $s_t$ shortly to show how it enables an explicit rate matrix. Since $(x_t, s_t)$ is governed by the sampled unmasking and insertion times, we write $(x_t, s_t) \sim p_t(\cdot \mid x_1)$. Marginalizing $p_t(\cdot \mid x_1)$ over $x_1 \sim p_1$ yields $p_t$. Since the boundary condition sets $\alpha_0 = \beta_0 = 0$, all tokens are deleted at $t = 0$; $p_0$ is the point mass on the empty string.

**FlexMDM training.** We now explain how we train our FlexMDM to learn the desired rate matrix. We first discuss what the CTMC looks like at a high level: recall from (6) that when $t$ *increases*, tokens are progressively inserted and unmasked. Indeed, one can show that a CTMC that generates the interpolant can be characterized by two quantities that govern the rate of insertion and unmasking:

- Unmasking posterior (modeled by $f_\theta(x, t)[i] \in \Delta(\Sigma)$): for each index $i$ that $x^i = \mathbf{m}$, the posterior distribution over the underlying clean token.
- Insertion expectation (modeled by $g_\theta(x, t)[i] \in \mathbb{R}_{\geq 0}$): for all indices $i$ in $x$, the expected number of tokens that remain to be inserted in between $x^{i-1}$ and $x^i$.

$f_\theta$ resembles the familiar unmasking posterior from MDMs, whereas $g_\theta$ is new: it predicts how many tokens need to be inserted. Intuitively, modeling a *variable-length* $p_1$ is harder than the fixed-length setup of MDM–introducing an insertion expectation allows us to parameterize more complicated CTMC for FlexMDM; its rate matrix will appear soon in Proposition 2. To define the training loss, we set the boundary values of $s_t$ as $s_t[-1] \coloneqq -1$ and $s_t[\mathrm{len}(s_t)] \coloneqq \mathrm{len}(x_1)$, and let $\phi(x, y) = y - x - x \log \frac{x}{y}$ denote a scalar Bregman divergence.

$$\mathcal{L}_\theta = -\int_0^1 \mathbb{E}\left[ \underbrace{\frac{\dot{\beta}_t}{1 - \beta_t} \sum_{x_t^i = \mathbf{m}} \log f_\theta(x_t, t)[i, x_1^{s_t[i]}]}_{\text{unmasking loss}} + \underbrace{\frac{\dot{\alpha}_t}{1 - \alpha_t} \sum_{i=0}^{\mathrm{len}(x_t)} \phi(s_t[i] - s_t[i-1] - 1, g_\theta(x_t, t)[i])}_{\text{insertion loss}} \right] dt \tag{7}$$

---

[1] Writing $x_t^i$ to mean the symbol derived from source position $i$ is this a mild abuse of notation since the superscript $i$ refers to a position in $x_1$ rather than a valid index of the (shorter) sequence $x_t$.

Here, the expectation is taken over $x_1 \sim p_1$, $(x_t, s_t) \sim p_t(\cdot|x_1)$. Proposition 1 exactly characterizes the unmasking posterior and insertion expectation and shows they uniquely minimize (7).

**Proposition 1** (FlexMDM training loss). *The loss $\mathcal{L}_\theta$ in (7) is uniquely minimized at*

$$f_\theta(x,t)[i,v] = \underbrace{\mathbb{P}(x_1^{s_t[i]} = v | x_t = x)}_{\text{unmasking posterior}}, \quad g_\theta(x,t)[i] = \underbrace{\mathbb{E}[s_t[i] - s_t[i-1] - 1 | x_t = x]}_{\text{insertion expectation}}.$$

These quantities match the explanation above: the posterior over the clean token together with the expected number of insertions. They precisely determine the FlexMDM rate matrix stated next.

**Proposition 2** (FlexMDM Rate Matrix). *Let the rate matrix $R_t$ be defined as:*

$$\begin{aligned} \textit{Unmask} : R_t\left(x, x[x^i \leftarrow v]\right) &= \tfrac{\dot{\beta}_t}{1-\beta_t} \cdot \mathbb{P}(x_1^{s_t[i]} = v | x_t = x), \quad v \in \Sigma, x^i = \mathbf{m} \\ \textit{Insert} : R_t\left(x, x \lhd_i \mathbf{m}\right) &= \tfrac{\dot{\alpha}_t}{1-\alpha_t} \cdot \mathbb{E}\left[s_t[i] - s_t[i-1] - 1 | x_t = x\right], \end{aligned} \tag{8}$$

*where $x \lhd_i \mathbf{m}$ is the sequence obtained from $x$ by inserting a mask token in between $(x^{i-1}, x^i)$. Then $R_t$ solves the KFE (equation (3)) with $p_t$ as the probability mass function of the FlexMDM interpolant $x_t$.*

Proposition 1 thus implies that minimizing the loss yields exact recovery of the rate matrix. In practice, we could simulate the CTMC using the learned networks $(f_\theta, g_\theta)$ in place of the ground-truth quantities in (8). By denoting the resulting terminal distribution as $p_1^\theta$, the variational loss quantifies the terminal-time KL divergence:

$$\mathcal{D}_{\text{KL}}(p_1 || p_1^\theta) \leq \mathcal{L}_\theta$$

We defer formal demonstration of propositions and the KL divergence guarantee to Appendix D. Definition of the joint interpolant is reinstated in definition D.2, the rate matrix in proposition D.3, the loss and variational bound in proposition D.4.

**Remark.** Our FlexMDM interpolant introduces only one extra quantity beyond MDM's unmasking posterior: the insertion expectation, a simple scalar per position. This stems from our design choice to gradually insert and then unmask a token. As shown in Section 5.2, this enables efficient task transfer of pretrained MDM weights. In contrast, alternative interpolants would require modeling more complex objects, such as a full token distribution, adding unnecessary training burden.

## 4 VARIABLE LENGTH MASKED DIFFUSIONS: INFERENCE

In this section, we outline inference algorithms for FlexMDM, focusing on two variants: **vanilla inference** and **adaptive inference**. We begin with a brief overview of inference in MDMs.

**Adaptive inference in MDM.** For the case of MDM, MDM inference proceeds by simulating the rate matrix entries in 5. From a high-level one way this can be done is by (a) independently sampling a subset of masked tokens to unmask and (b) sampling clean tokens from the unmasking posterior. Crucially for what follows, the same guarantee holds for non-independent *adaptive* choices of unmasking indices, e.g., confidence-based: correctness hinges on using the ground-truth unmasking posterior, not on following the rate matrix's unmasking entries. This adaptive inference strategy is widely used due to its empirical performance. We adopt this template and show that FlexMDM inherits the same any-order property.

**Vanilla inference.** We begin with the *vanilla inference* of FlexMDM, which is obtained by discretizing the CTMC in (8) using trained neural networks $(f_\theta, g_\theta)$. Choosing an appropriate discretization scheme is crucial, as different schemes can lead to markedly different empirical behavior. We adopt $\tau$-*leaping*—originating in chemical physics and shown to outperform naive Euler discretization for MDMs (Campbell et al., 2022)—which batches all events occurring within a fixed interval $[t, t+\tau]$. At a high level, for each discretized step, we simultaneously (Figure 3, right):

- Unmasking: For each mask token, sample for every unmasking a number according to the unmasking intensities in the rate matrix. Unmask only if a non-zero entry is returned.
- Insertion: Sample the number of *mask-token insertions* from a Poisson distribution parameterized by the insertion rate, then apply those insertions.

---

**Subroutine 1:** VLMDM inference

**Require:** Learned functions $(f_\theta, g_\theta)$
**Require:** Discretization $0 = t_1 < \cdots < t_N = 1$
**Require:** Insertion, Unmasking schedule $\alpha_t, \beta_t$
1: Initialize $X_{t_1} \leftarrow \varepsilon$
2: **for** $k = 1$ to $N - 1$
3:     $\tau \leftarrow t_{k+1} - t_k$
4:     **Invoke Subroutine 2 for unmasking**
5:     **for** $i$ in $[\text{len}(X_{t_k})]$
6:        Set rate $r \leftarrow \frac{\dot{\alpha}_{t_k}}{1 - \alpha_{t_k}} \cdot \tau$
7:        Sample $\ell \sim \text{Poi}\left(r \cdot g_\theta(X_{t_k}, t_k)[i]\right)$
8:        **Insert $\ell$ masks between $X_{t_k}^{i-1}$ and $X_{t_k}^i$**
9: **return** $X_{t_N}$

---

**Subroutine 2:** Unmasking Step

1: **if vanilla inference** :
2:     **for** $i \in \{i | X_{t_k}^i = \mathbf{m}\}$ and $v \in \Sigma$
3:        Set rate $r \leftarrow \frac{\dot{\beta}_{t_k}}{1 - \beta_{t_k}} \cdot \tau$
4:        $k_v \sim \text{Poi}(r \cdot f_\theta(X_{t_k}, t_k)[i, v])$
5:        **if** $^{\exists!}v$ such that $k_v = 1$
6:           Set $X_{t_k}^i \leftarrow v$
7: **if adaptive inference** :
8:     Select K (the size of $|S|$)
9:     **for** $i \in \{i | X_{t_k}^i = \mathbf{m}\}$
10:      Compute confidence $\mathcal{C}^i$
11:     **for** $i$ in $\text{argmaxK}(C)$
12:      $X_{t_k}^i \sim \text{Cat}(f_\theta(X_{t_k}, t_k)[i])$

---

**Algorithm 1: VLMDM inference.** At each step we perform **unmasking** and **insertion**. For **unmasking**, unmask by $\tau$-leaping (**vanilla**) or by confidence-based selection (**adaptive**). The number of mask tokens to **insert** is drawn from a Poisson distribution. **Notation**: Cat, Poi imply the categorical and Poisson distribution, respectively. $\text{argmaxK}(\mathcal{C})$ is the indices set of the $K$ largest components of $\mathcal{C}$. We provide more details in Appendix E.

As the number of steps $\to \infty$, this inference algorithm recovers the CTMC and the discretization error vanishes. Algorithm 1 details the full sampler.

**Adaptive inference.** Notably, one can choose the positions to unmask adaptively. Precisely, at each inference step we select the unmasking positions according to a heuristic rule that prioritizes the most confident indices, where confidence is computed either from the *model's unmasking posterior* or via a *semi-autoregressive* rule (prioritizing leftmost masks). We find such adaptive choice substantially boosts performance; see Section 5. Since unmasking indices in an adaptive no longer trace the transitions described by the rate matrix entries defined in (8), one might ask whether sampling still guarantees to sample from the target distribution $p_1$ in the infinitesimal limit. The following proposition answers in the affirmative.

**Proposition 3** (Any-order inference, informal). *Consider any sampling scheme that, at each step: (i) unmasks an arbitrary subset of masked positions but draws revealed tokens from the* ground-truth unmasking posterior*; and (ii) applies* insertion *CTMC governed by the ground-truth rate matrix. Then the resulting process samples from the target distribution* $p_1$.

The formal statement and the proof of Proposition 3 are given in Appendix E. In words, following the unmasking entries of the rate matrix corresponding to the schedule used in training is *not* necessary to preserve the sampling guarantee. Moreover, the samplers as $N \to \infty$ in Algorithm 1 is subsumed by the class in Proposition 3, therefore, assuming access to the ground-truth unmasking posterior and insertion expectation, the corresponding class of algorithms in Algorithm 1 samples from $p_1$ up to discretization error.

**Remark.** A key technical ingredient underlying the rigor of our adaptive inference is that the respective entries of the unmasking posterior of the ground truth rate matrix in Proposition 3 do *not* depend on the choice of unmasking schedule $\beta_t$ (the proof is given in Appendix E.2.1). This independence allows a single model $f_\theta$ to learn all possible unmasking transitions arising along different paths that ultimately connect $p_0$ to $p_1$, thereby enabling adaptive unmasking at inference time. This feature is the same mechanism enabling adaptive inference for MDMs, but for FlexMDMs, proving that it interfaces correctly with insertions is quite subtle. Note that a similar notion—independence of the choice of path—has been introduced in continuous spaces Albergo et al. (2023a); Negrel et al. (2025). We defer further discussion to Appendix E.

## 5 EXPERIMENT

In this section, we present experimental results for FlexMDM, demonstrating the following:

- FlexMDM is an effective variable-length learner: length modeling, planning, local edits.

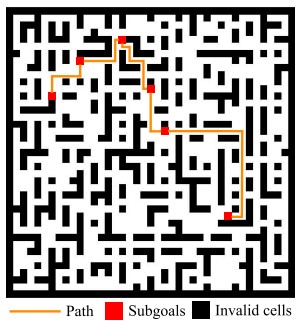 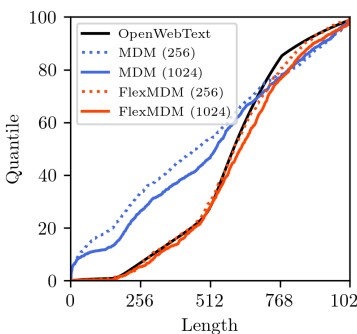 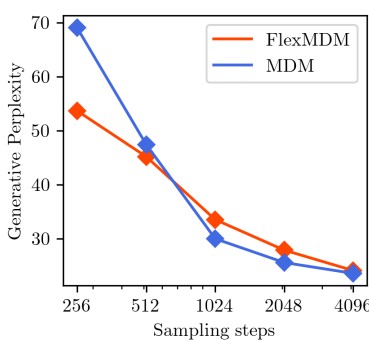

**(a) Maze task illustration.** The model is given subgoals and is required to connect them.

**(b) Length modeling.** FlexMDM recovers the true length distribution of OpenWebText training data.

**(c) Perplexity.** FlexMDM achieves generative perplexity on par with MDM.

- FlexMDM is scalable: 8B FlexMDM is obtainable by initializing from a pretrained MDM.

Section 5.1 presents from-scratch results for FlexMDM on text and planning tasks distributions, confirming its practical efficiency. Next, Section 5.2 provides an 8B-scale FlexMDM's training recipe, initialized from LLaDA-8B Nie et al. (2025), and evaluates it in math and code infilling tasks. We begin with the architectural and scheduling choices used throughout.

**Training design.** Recall from Section 3 that FlexMDM models the unmasking posterior $f_\theta$ and insertion expectation $g_\theta$ given state $x$ and time step $t$. We adopt DiT (Peebles & Xie, 2023), a bidirectional transformer that enables additional embedding, as a backbone. To learn both quantities jointly, we attach two output heads: a standard posterior head for $f_\theta$ and a scalar softplus head for $g_\theta$. Moreover, we choose our unmasking and insertion schedule to be both linear, $\alpha_t = \beta_t = t^2$.

## 5.1 PRETRAINING

In this section, we evaluate FlexMDM's ability to learn variable-length data from scratch. Our baseline is MDM, which is fixed-length but can handle variable-length sequences by padding to a fixed maximum length with an auxiliary pad token. This padding setup is widely used in instruction fine-tuning when variable-length answers are desired (Nie et al., 2025; 2024; Ye et al., 2025; Gong et al., 2024). For a fair comparison, we use vanilla inference for both MDM and FlexMDM throughout. Further experimental details appear in Appendix F.

### 5.1.1 PRETRAINING ON TEXT DATA

We first construct a training dataset from the raw OpenWebText corpus (Gokaslan et al., 2019), splitting each article into paragraphs to preserve semantic coherence and yield variable-length sequences. Models pretrained on this data, therefore, generate variable-length text.

**Results.** We train 175M FlexMDM and MDM with a maximum sequence length 1024 for 500K iterations and batch size 1024. Using the pretrained models, we vary the number of sampling steps and measure (a) generative perplexity as a proxy for text fluency, and (b) the induced length distribution. Figure 4c shows comparative generative perplexity for the two models, improving as sampling steps increase, indicating no fluency degradation for FlexMDM despite its more involved loss objective. Crucially, we observe that FlexMDM matches the true length distribution far more closely (Figure 4b): with only 256 steps it tracks the ground truth distribution (red line, whereas MDM remains miscalibrated even at 1024 steps (blue line). In Appendix F.1.1, we provide additional experimental results: the entropy of given sequences to ensure the generative perplexity is measured under similar conditions and the perplexities for larger sampling steps (8192, 16384, 32768).

**Remark.** We remark that our pretraining pipeline differs from prior MDM setups that truncate the corpus to a fixed maximum length. Also, one might ask why we do not provide additional metrics on text benchmarks, such as validation perplexity. This is because MDM and FlexMDM use different objectives (see equation (4) and equation (7)), making likelihood comparisons hard to interpret.

---

[2]The ground-truth unmasking posterior is independent of $\beta_t$, so we condition the network on $\alpha_t$ only; under the linear choice $\alpha_t = t$, this coincides with the usual time embedding.

Albeit, we experimentally confirm that both loss functions can serve as a reliable proxy of the likelihood, although with different scales, by evaluating both pretrained models on a downstream text benchmark (Appendix F.1.1). We address the concern about the absence of the metric by evaluating scaled models on downstream benchmarks in Section 5.2.

### 5.1.2 PLANNING TASK

We further evaluate FlexMDM's ability in a planning task in a discrete space. Motivated by an earlier study Janner et al. (2022) that investigated the ability of continuous diffusion in maze tasks, we design a grid-maze benchmark: the maze is fixed but unknown to the model, with a subset of cells invalid. Given a sequence of subgoal grids $(g_1, \ldots, g_K)$, the model must connect this sequence without entering invalid cells (see Figure 4a). This subgoal structure aligns naturally with FlexMDM: starting from $(g_1, \ldots, g_K)$, inference inserts mask tokens between subgoals and then unmasks to generate a feasible path. Theoretically, this can be seen as augmenting the base distribution to contain a data-dependent distribution Albergo et al. (2024). This is in stark contrast to MDM, where it must preassign each subgoal to a specific position, which is difficult to know *a priori*. We provide additional details in Appendix F.2.

**Results.**    We use a $41 \times 41$ maze and control the task's difficulty via varying the number of subgoals $K \in \{2, 7, 12\}$. As $K$ increases, MDM performance degrades markedly, while FlexMDM maintains robust success rates, reaching a gap of up to 60% at $K = 12$. These results firmly support FlexMDM as a principled approach for subgoal-based planning, where preallocating token positions is inherently challenging for fixed-length models.

| Difficulty | MDM | FlexMDM |
|---|---|---|
| Easy | 68.4% | **92.3%** |
| Medium | 29.3% | **90.4%** |
| Hard | 24.2% | **90.0%** |

Table 1: FlexMDM outperforms MDM on the subgoal-style maze-planning task.

### 5.2 SCALING UP FLEXMDM

In this section, we address FlexMDM's scalability by scaling it to 8B parameters and observing notable improvements over an MDM baseline. We start from the observation that MDM and FlexMDM both share the unmasking posterior as a core component, suggesting effective *task transfer* from a pretrained MDM might be possible. To demonstrate this, concretely, we initialize from LLaDA-Base (Nie et al., 2025) and make the following modifications: (a) add time-embedding layers and a scalar head to model the insertion expectation; (b) attach LoRA adapters. Altogether, the resulting number of trainable parameters is ≈400M. To cover both natural and mathematical language, we train on the 50:50 mixture of OpenWebText (Gokaslan et al., 2019) and Proof-Pile-2 (Azerbayev et al., 2023). Surprisingly, we observe rapid transfer: within three days on 16 H100 GPUs, the model generates variable-length sentences. Notably, the number of tokens used is ≈13.1 B, and this is in contrast much smaller than the number of LLaDA-Base pretraining tokens (≈1.5T). We then instruction-fine-tune (IFT) this base FlexMDM to evaluate it on downstream tasks. See Appendix F for more details. [3]

**Results.** For comparison, we train FlexMDM and LLaDA-Base from the same number of IFT pairs. For math and code, respectively, we IFT on the GSM8K train split(Cobbe et al. (2021); ≈8000 pairs) and the educational split of opc-sft-stage-2 (Huang et al., 2024) (≈0.1M pairs), for which IFT-ed models are evaluated in the GSM8K test split and HumanEval-infill (single line) (Bavarian et al., 2022) in zero-shot. Sampling is done by confidence-based sampling with a sliding window. Notably, as the number of sampling steps increases, FlexMDM continues to improve, highlighting its strength in reasoning tasks given sufficient compute—whereas the IFT-ed LLaDA's performance remains flat. Although in this experiment we use IFT on task-specific pairs, we expect that training on a much more diverse instruction–answer pairs with sufficient compute will yield a more generalized model.

---

[3] For a fair comparison, since FlexMDM is not IFT-ed, we IFT LLaDA-Base, rather LLaDA-instruct, this differs from Zhao et al. (2025). We employ zero-shot evaluation, which also differs from Nie et al. (2024).

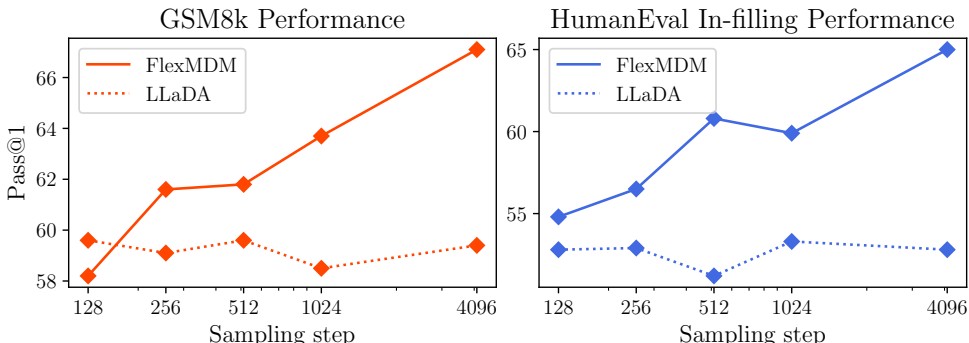

**Figure 5:** FlexMDM performance exhibits superior scaling when more sampling steps are allocated.

## 6 CONCLUSION

In this work we proposed Flexible Masked Diffusion Models (FlexMDM), a discrete diffusion framework over variable-length sequences. Theoretically, via a joint interpolant viewpoint, we provide rigorous guarantees for both training and inference of FlexMDM. Empirically, FlexMDM learns variable-length structure across diverse scenarios, scales to 8B parameters, trains in only a few GPU-hours, and yields substantial improvements on math and coding infilling tasks. Further exploration of FlexMDM's capabilities is a promising direction for future work.

Beyond these results, our goal is to align generative modeling with how humans and nature compose discrete sequences. Instead of filling fixed positions; they **insert**, revise, and **reorder** tokens. We hope that our work takes a step in this direction.

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

## CONTENTS

## A    RELATED WORKS

**Discrete diffusion and flows.**    Early diffusion models were formulated as continuous-time Markov chains over continuous spaces with Gaussian transition kernels (Sohl-Dickstein et al., 2015; Ho et al., 2020), and were later connected to continuous-time formulations via stochastic differential equations, offering a unifying perspective on score-based generative modeling (Song et al., 2020). In parallel, *discrete* diffusion has been developed from the viewpoint of Markov chains over discrete space (Hoogeboom et al., 2021). Notably, Austin et al. (2021) introduced D3PM with several families of discrete transition kernels, and Lou et al. (2023) proposed SEDD, which adopts score-based training objectives. A complementary line of work studies *discrete flows* (Campbell et al., 2024; Gat et al., 2024), aiming to understand continuous-time Markov chains (CTMCs) that interpolate between data and base distributions; this perspective aligns with ours. Subsequent extensions consider token-wise paths and path-wise structure within such flows (Shaul et al., 2024).

**Masked Diffusion Models.**    Among discrete-transition designs, absorbing-state (a.k.a. masking) kernels have become a popular and strong-performing choice. Recent work shows that this yields a simple and principled training recipe, referred to as Masked Diffusion Models (MDMs) (Sahoo et al., 2024; Shi et al., 2024). A growing body of results demonstrates the scalability of this approach across problem settings and modalities, including large-scale natural language modeling (Nie et al., 2024; 2025; Ye et al., 2025; Song et al., 2025; DeepMind, 2025), code generation (Labs et al., 2025; Gong et al., 2025), and multimodal learning (Swerdlow et al., 2025).

**Any-order inference in MDMs.**    With the advent of MDMs, subsequent work has established that they admit theoretically grounded *any-order inference*, wherein tokens can be unmasked in arbitrary orders rather than following a fixed CTMC schedule (Kim et al., 2025; Peng et al., 2025). Practical token-ordering rules span a spectrum of heuristics based on model confidence and uncertainty—e.g., maximum-probability logits (Chang et al., 2022; Zheng et al., 2024), probability margin (Kim et al., 2025), semi-autoregressive schedules (Nie et al., 2024), and entropy-based criteria (Ben-Hamu et al., 2025)—as well as strategies that leverage reference models to guide the unmasking trajectory (Peng et al., 2025). Beyond heuristics, another thread trains auxiliary modules to anchor or adapt the generation order (Rout et al., 2025), while recent work directly *learns* token orders end-to-end (Ma et al., 2025a; Wang et al., 2025).

**Stochastic interpolant.**    Stochastic interpolant (Albergo & Vanden-Eijnden, 2022; Albergo et al., 2023b) is a general framework for building measure-transport based generative models on continuous state space. While building off different philosophical grounds, it can be seen as equivalent to flow matching (Lipman et al., 2022; Liu et al., 2022). Extensions of the interpolant have been proposed for conditional generation through data-dependent coupling (Albergo et al., 2024), which we adopt for infilling task design in Section 5.

**Descriptive overview on concurrent work.**    The most notable concurrent work is Edit-Flow (Havasi et al., 2025), where the primary mathematical machinery that enabled their construction is referred to as "Flow Matching with Auxiliary Process". We note that this can be interpreted as mathematically equivalent to the notion of joint interpolant in this work, e.g., our Proposition D.1 is equivalent to Theorem 3.1 in Havasi et al. (2025).

The main differences are (1) the choice of interpolant and (2) the guarantee of any-order inference. Whereas EditFlow is built around an explicit probability path, we instead define a pair of coupled random variables that implicitly induce this path, leading to a different choice of intermediate. As discussed in Section 3, our choice of interpolant yields a distinct training objective for an unmasking posterior and an insertion-expectation term. Consequently, it enables the any-order inference guarantee established in Section 4.

## B    NOTATION

In this section, we reiterate the notations used in the main body and introduce auxiliary notations that are used in the proofs of the appendix.

**Strings.** Let $\varepsilon$ denote the empty string, $\Sigma$ a vocabulary of words, $\mathbf{m}$ a special mask token. We write $x^i$ the $i$-th element of $x$ with 0-baesd indexing, $x|_S$ the string indexed by an index set, e.g. $abc[\{0, 2\}] = ac$. To insert a token $v$ before position $i$ in string $x$, we write $x \triangleleft_i v$, e.g., to prepend a token $abc \triangleleft_0 d = dabc$ and to append a token $abc \triangleleft_3 d = abcd$. To replace the $i$-th token in $x$ with $v$, we write $x[x^i \leftarrow v]$. As much of the work involves masking, we write $x \subseteq y$ if $x$ can be constructed by partially masking $y$. We write $\mathrm{mask}(x), \mathrm{unmask}(x) \subseteq [\mathrm{len}(x)]$ for the set of indices corresponding to mask and clean tokens.

## C  DISCRETE STOCHASTIC INTERPOLANTS: DEFINITIONS AND PROPOSITIONS FOR SECTION 2

In continuous spaces, a common approach to define generative transport is the stochastic interpolant framework, which implicitly defines the interpolation distribution $p_t$ by specifying an interpolant $\{x_t\}_{t \in [0,1]}$ and regressing the required quantities to realize the transport.

In the section C.1, we introduce a discrete analogue of the stochastic interpolant. To illustrate this framework, we reformulate the widely used masked diffusion model for sequences of length $n$ within our setup. Briefly, masked diffusion defines the interpolation $p_t$ by progressively unmasking tokens in sentences drawn from the data distribution. At time $t = 0$, all tokens are masked, so $p_0$ is a point mass at the fully masked sequence, the $\mathbf{m}$ token repeated $n$ times. The transition rates driving the generative transport can be characterized as functions of per-token posterior probabilities conditioned on time $t$. These are typically learned by minimizing a variational objective in the form of a weighted cross-entropy loss.

### C.1  DISCRETE STOCHASTIC INTERPOLANT

To obtain the target rate matrix, the discrete stochastic interpolant relies on an interpolating rate matrix that drives a sample from a sample drawn from a sample from $p_0$ to a sample from $p_1$, defined as follows:

**Definition C.1** (Discrete Stochastic Interpolant and Interpolating Rate). *Let $x_0 \sim p_0$ and $x_1 \sim p_1$. A **discrete stochastic interpolant** is a family of random variables $\{x_t\}_{t \in [0,1]}$, defined on a common probability space and satisfying the boundary conditions $x_{t=0} = x_0$ and $x_{t=1} = x_1$, for which there exists a continuous-time Markov chain with bounded, time-dependent transition rate matrix $K_t^{x_0,x_1}$ such that, for each $t \in [0, 1]$, $\mathrm{Law}(x_t \mid x_0, x_1)$ coincides with the marginal distribution at time $t$ of that Markov chain started at $X_0$. We refer to $K_t^{x_0,x_1}$ as an **interpolating rate matrix**.*

With an interpolating rate of an interpolant, the target rate matrix can then be obtained through Proposition C.1

**Proposition C.1** (Target Rate). *Given a discrete stochastic interpolant $x_t$ and an interpolating rate matrix $K_t^{x_0,x_1}$, the continuous-time Markov chain with initial distribution $p_0$ and target transition rate matrix $R_t$ defined as,*

$$R_t(x, y) = \mathbb{E}_{x_0, x_1}[K_t^{x_0,x_1}(x, y)|x_t = x]$$

*has marginals equal to $\mathrm{Law}(x_t)$.*

*Proof.* Writing $p_t$ a probability mass function (pmf) of $x_t$ and $p_t(\cdot|x_0, x_1)$ a pmf of $x_t$ conditioned on $x_0, x_1$. We further write $q(x_0, x_1)$ the joint pmf of $x_0$ and $x_1$ and $q_t(x_0, x_1|x_t)$ to be the joint pmf conditioned on $x_t$.

It suffices to show $R_t$ satisfies the Kolmogorov Forward Equation with pmf $p_t$ as follows:

$$
\begin{aligned}
\sum_y R_t(y,x)p_t(y) &= \sum_y \mathbb{E}_{x_0,x_1}[K_t^{x_0,x_1}(y,x)|x_t=y]p_t(y) \\
&= \sum_y \sum_{x_0,x_1} K_t^{x_0,x_1}(y,x)q_t(x_0,x_1|y)p_t(y) \\
&= \sum_y \sum_{x_0,x_1} K_t^{x_0,x_1}(y,x)p_t(y|x_0,x_1)q(x_0,x_1) \\
&= \mathbb{E}_{x_0,x_1}\left[\sum_y K_t^{x_0,x_1}(y,x)p_t(y|x_0,x_1)\right] \\
&= \mathbb{E}_{x_0,x_1}[\partial_t p_t(y|x_0,x_1)] \\
&= \partial_t p_t(x)
\end{aligned}
$$

This concludes the proof. $\qquad\square$

**Remarks.** While written considerably differently, the framework is mathematically equivalent to discrete flow matching Gat et al. (2024). The difference is only philosophical: discrete flow matching relies on the notion of a conditional probability path that the interpolating rate should induce, whereas we define such a probability path only implicitly through the definition of the interpolant.

## C.2 THE MASKED DIFFUSION INTERPOLANT

As a concrete example of the discrete stochastic interpolant, we reformulate the masked diffusion model and its learning in the framework. As masked diffusion starts from a point mass, we drop the dependence of $x_0$ in writing.

**Definition C.2** (The Masked Diffusion Interpolant). *Let $x_1 \sim p_1$ be a sentence of length $n$ drawn from the data and $\alpha_t$ a smooth unmasking schedule that interpolates from $\alpha_{t=0} = 0$ to $\alpha_{t=1} = 1$. Define the unmasking times $\{T^i\}_{i \in [0,\dots,n-1]}$ as:*

$$
\forall i \in \{0,\dots,n-1\}: \quad T^i \sim \dot{\alpha}_t\, dt,
$$

*Then, the masked diffusion interpolant is defined as:*

$$
x_t = \begin{cases} \mathbf{m} & \text{if } t < T_i, \\ x_1^i & \text{if } t \geq T_i. \end{cases}
$$

In other words, at each time $t$, $x_t$ reveals a subset of the tokens of $x_1$, with each token $x_1^i$ independently unmasked at its associated time $T^i$.

**Proposition C.2** (The Masked Diffusion Interpolating Rate). *One interpolating rate $K_t^{x_1}$ of the masked diffusion interpolant $x_t$ is given by:*

$$
\forall x \subseteq x_1, v \in \Sigma, x^i = \mathbf{m}: \quad K_t(x, x[x^i \leftarrow v]) = \frac{\dot{\alpha}_t}{1-\alpha_t}\mathbf{1}\{v = x_1^i\}.
$$

*Proof.* Let $p_t(\cdot|x_1)$ as the pmf of $x_t$ conditioned on $x_1$. From the definition of the interpolant, we notice that:

$$
p_t(x|x_1) = \prod_{i=0}^{\text{len}(x_1)-1} \left[(1-\alpha_t)\mathbf{1}\{x^i = \mathbf{m}\} + \alpha_t\mathbf{1}\{x^i = x_1^i\}\right]
$$

We verify that $K_t$ satisfies the Kolmogorov Forward Equation (3) under the conditioned pmf as follows,

L.H.S

$$= \partial_t p_t(x|x_1)$$

$$= \partial_t \left[ \prod_{i=0}^{\text{len}(x_1)-1} (1-\alpha_t)\mathbf{1}\{x^i = \mathbf{m}\} + \alpha_t \mathbf{1}\{x^i = x_1^i\} \right]$$

$$= \sum_{i=0}^{\text{len}(x_1)-1} \left( -\dot{\alpha}_t \mathbf{1}\{x^i = \mathbf{m}\} + \dot{\alpha}_t \mathbf{1}\{x^i = x_1^i\} \right) \cdot \prod_{j \neq i} (1-\alpha_t)\mathbf{1}\{x^j = \mathbf{m}\} + \alpha_t \mathbf{1}\{x^j = x_j^i\},$$

R.H.S

$$= \sum_{i=0}^{\text{len}(x_t)-1} \mathbf{1}\{x^i = x_1^i\} \frac{\dot{\alpha}_t}{1-\alpha_t} p_t(x[x^i \leftarrow \mathbf{m}]|x_1) - \mathbf{1}\{x^i = \mathbf{m}\} \frac{\dot{\alpha}_t}{1-\alpha_t} p_t(x|x_1)$$

$$= \sum_{i=0}^{\text{len}(x_t)-1} \mathbf{1}\{x^i = x_1^i\} \frac{\dot{\alpha}_t}{1-\alpha_t}(1-\alpha_t) \prod_{j \neq i} (1-\alpha_t)\mathbf{1}\{x^j = \mathbf{m}\} + \alpha_t \mathbf{1}\{x^j = x_j^i\}$$

$$- \sum_{i=0}^{\text{len}(x_t)-1} \mathbf{1}\{x^i = \mathbf{m}\} \frac{\dot{\alpha}_t}{1-\alpha_t}(1-\alpha_t) \prod_{j} (1-\alpha_t)\mathbf{1}\{x^j = \mathbf{m}\} + \alpha_t \mathbf{1}\{x^j = x_j^i\})$$

$$= \text{L.H.S.}$$

This concludes the proof. $\square$

Note that since each $T^i$ is sampled independently from a continuous distribution, the probability that two unmasking times coincide is zero. Thus, only a single token is unmasked in any infinitesimal transition almost surely.

**Proposition 4** (The Masked Diffusion Target Rate). *By proposition C.1, a target rate $R_t$ that induces Law$(x_t)$ is:*

$$\forall x \subseteq x_1, v \in \Sigma, x^i = \mathbf{m}: \quad R_t(x, x[x^i \leftarrow v]) = \frac{\dot{\alpha}_t}{1-\alpha_t} \mathbb{P}(x_1^i = v | x_t = x).$$

*Proof.* Following proposition C.2, the result follows from invoking proposition C.1. $\square$

To learn an approximation to $R_t$, we now parameterize an approximate target rate of the form $\hat{R}_t(x, x[x^i \leftarrow v]) := \frac{\dot{\alpha}_t}{1-\alpha_t} f_\theta(x, t)[i, v]$ where $f_\theta(x, t)[i, v]$ is a learned approximation to the posterior $\mathbb{P}(x_1^i = v \mid x_t = x)$.

The target rate can then be characterized by a variational objective that measures the discrepancy between the true and approximate path measures.

**Proposition 5** (Variational Loss for Masked Diffusion). *The loss function is defined as:*

$$L[\hat{R}_t] = \int_0^1 \mathbb{E}_{x_1, x_t} \left[ -\frac{\dot{\alpha}_t}{1-\alpha_t} \sum_{i=0}^{n-1} \mathbf{1}\{x_t^i = \mathbf{m}\} \log f_\theta(x_t, t)[i, x_1^i] \right] dt,$$

*is uniquely minimized when $\hat{R}_t = R_t$, and is connected to the terminal KL-divergence by:*

$$\mathcal{D}_{KL}(p_1||\hat{p}_1) \leq L[\hat{R}_t],$$

*where $\hat{p}_1$ is the approximate data distribution generated by $\hat{R}_t$.*

*Proof.* Let $\mathbb{P}$ and $\hat{\mathbb{P}}$ be the path measures associated with the continuous-time Markov chain of the target rate matrix $R_t$ in proposition C.1 and an approximation through a neural network $\hat{R}_t$. The

variational loss follows by expanding the KL-divergence between the two path measures.

$$
\begin{aligned}
\mathcal{D}_{\mathrm{KL}}(\mathbb{P} \parallel \hat{\mathbb{P}}) &= \mathbb{E}_{\mathbb{P}}\left[\int_{t=0}^{t=1} R_t(x_t, x_t) - \hat{R}_t(x_t, x_t)\, dt + \sum_{t:x_t \neq x_{t-}} \log \frac{R_t(x_{t-}, x_t)}{\hat{R}_t(x_{t-}, x_t)}\right] \\
&= \int_0^{t=1} \mathbb{E}_{x_t}\left[\sum_{y \neq x_t} \hat{R}_t(x_t, y) - \sum_{y \neq x_t} R_t(x_t, y) - R_t(x_t, y)\log \frac{R_t(x_t, y)}{\hat{R}_t(x_t, y)}\right] dt \\
&= \int_0^1 \mathbb{E}_{x_1, x_t}\left[-\frac{\dot{\alpha}_t}{1 - \alpha_t}\sum_{i=0}^{n-1} \mathbf{1}\{x_t^i = \mathbf{m}\}\log f_\theta(x_t, t)[i, x_1^i]\right] dt
\end{aligned}
$$

where the first line takes the expectation of the Radon-Nikodym derivative between the two path measures. The statement of Radon-Nikodym derivative between two CTMCs can be found in the Appendix in Campbell et al. (2024). A discrete-time equivalent derivation can also be found in Shaul et al. (2025).

The terminal KL bound then follows directly from the data processing inequality, that is:

$$
\mathcal{D}_{\mathrm{KL}}(p_1 \parallel \hat{p}_1) \leq \mathcal{D}_{\mathrm{KL}}(\mathbb{P} \parallel \hat{\mathbb{P}})
$$

This technique is standard, as shown in Vargas et al. (2025) for the case of path reversal-based construction of diffusion generative models, and in Holderrieth et al. (2025a) for discrete diffusion.

$\square$

## D  JOINT DISCRETE STOCHASTIC INTERPOLANTS: DEFINITIONS AND PROPOSITIONS FOR SECTION 3

Building on the discrete stochastic interpolant, we proceed to construct a discrete diffusion that models a probability distribution whose supports span variable-length sequences.

On a high level, we would like to define an interpolant constructed by deleting and masking sentences from the data distribution. However, the corresponding interpolating rate becomes cumbersome to characterize, as it is no longer clear what each mask token should unmask to.

To this end, we introduce the **joint interpolant** that allows us to construct a broader class of interpolants and interpolating rate matrices by augmenting the interpolant with auxiliary information that allows us to specify a more flexible interpolation path. We then leverage this newfound freedom to construct the flexible-length masked diffusion model.

### D.1  JOINT INTERPOLANT

By introducing an auxiliary variable coupled with the interpolant, the joint interpolant expands the class of interpolating rates that can be defined.

**Definition D.1** (Joint Interpolant and Joint Interpolating Rate). *Let $x_0 \sim p_0$ and $x_1 \sim p_1$. A **joint interpolant** is a family of coupled random variables $\{(x_t, s_t)\}_{t \in [0,1]}$ defined on a common probability space and satisfying the boundary conditions $x_{t=0} = x_0$ and $x_{t=1} = x_1$, for which there exists a continuous-time Markov chain with bounded, time-dependent transition rate matrix $K_t^{x_0, x_1}$ on the joint state space such that, for each $t \in [0,1]$, the conditional law $\mathrm{Law}(x_t, s_t \mid x_0, x_1)$ coincides with the marginal distribution at time $t$ of this Markov chain started at $(x_0, s_0)$. We call $K_t^{x_0, x_1}$ a **joint interpolating rate matrix**.*

**Proposition D.1** (Joint Interpolant Target Rate). *Let $\{(x_t, s_t)\}_{t \in [0,1]}$ be a joint interpolant with joint interpolating rate matrix $K_t^{x_0, x_1}$. Consider the continuous-time Markov chain with initial distribution $p_0$ and target transition rate matrix $R_t$ defined by*

$$
R_t(x, y) = \mathbb{E}_{s_t, x_0, x_1}\left[\sum_{s' \in \mathcal{S}} K_t^{x_0, x_1}\big((x, s_t), (y, s')\big) \mid x_t = x\right],
$$

*where $\mathcal{S}$ denotes the discrete state space of the auxiliary variable $s_t$. The marginal of the chain at time $t$ is then equal to $\mathrm{Law}(x_t)$.*

We note that this result is equivalent to "Flow Matching with Auxiliary Process" in concurrent work Havasi et al. (2025).

*Proof.* Let $p_t(\cdot, \cdot | x_0, x_1)$ the joint pmf of $x_t$, $s_t$ conditioned on $x_0, x_1$, let $q_t(x_0, x_1, s_t | x_t)$ to be the joint of $x_0, x_1, s_t$ conditioned on $x_t$, and let $q_t(x_0, x_1)$ to be the joint of $x_0, x_1$. We proceed to verify $R_t$ satisfies the KFE as in Equation 3,

$$
\begin{aligned}
\sum_y R_t(y, x) p_t(y) &= \sum_y \mathbb{E}_{s_t, x_0, x_1} \left[ \sum_{s' \in \mathcal{S}} K_t^{x_0, x_1} \big( (y, s_t), (x, s') \big) \mid x_t = y \right] p_t(y) \\
&= \sum_y \sum_{s_t, x_0, x_1} \sum_{s' \in \mathcal{S}} K_t^{x_0, x_1} \big( (y, s_t), (x, s') \big) q_t(x_0, x_1, s_t | y) p_t(y) \\
&= \sum_y \sum_{s_t, x_0, x_1} \sum_{s' \in \mathcal{S}} K_t^{x_0, x_1} \big( (y, s_t), (x, s') \big) q_t(x_0, x_1, s_t | y) p_t(y) \\
&= \sum_y \sum_{s_t, x_0, x_1} \sum_{s' \in \mathcal{S}} K_t^{x_0, x_1} \big( (y, s_t), (x, s') \big) p_t(y, s_t | x_0, x_1) q(x_0, x_1) \\
&= \mathbb{E}_{x_0, x_1} \left[ \sum_{s'} \partial_t p_t(x, s' | x_0, x_1) \right] \\
&= \partial_t p_t(x | x_0, x_1)
\end{aligned}
$$

This concludes the proof. $\qquad\square$

### D.2 FLEXIBLE-LENGTH MASKED DIFFUSION

We then instantiate the joint interpolant to obtain the length-aware masked diffusion model, using a sorted list of indices that has been inserted. Again, we drop $x_0$ in the writing as the model interpolates between a point mass at an empty sentence to the full data distribution. We redefine the interpolant in equation 6 for clarity.

**Definition D.2** (Flexible-Length Masked Diffusion Joint Interpolant). *Let $x_1 = (x_1^0, \ldots, x_1^{n-1}) \sim p_1$ be a sequence of length $n$. Let $\alpha_t$ and $\beta_t$ be monotone and differentiable schedules on $[0, 1]$ such that $\alpha_0 = \beta_0 = 0$ and $\alpha_1 = \beta_1 = 1$. Define insertion and unmasking times $\{T_1^i\}_{i=0}^{n-1}$, $\{T_2^i\}_{i=0}^{n-1}$ as follows:*

$$T_1^i \sim \dot{\alpha}_t \, dt,$$

$$T_2^i \sim \mathbf{1}\{t \geq T_1^i\} \cdot \frac{\dot{\beta}_t}{1 - \beta_{T_1^i}} \, dt.$$

*At each time $t \in [0, 1]$, define the sorted index set $s_t$ as:*

$$s_t = \{i \in \{0, \ldots, n-1\} \mid t > T_1^i\},$$

*with ascending order $s_t[0] < \cdots < s_t[\mathrm{len}(s_t) - 1]$, and boundary values:*

$$s_t[-1] = -1, \quad s_t[\mathrm{len}(s_t)] = n,$$

*and define the interpolant state $x_t$ per-coordinate as:*

$$
x_t^i = \begin{cases} \mathbf{m} & \text{if } t < T_2^{s_t[i]}, \\ x_1^{s_t[i]} & \text{if } t \geq T_2^{s_t[i]}. \end{cases}
$$

*The process $(s_t, x_t)_{t \in [0, 1]}$ is the **flexible length masked diffusion joint interpolant**.*

Here, $s_t$ tracks the ordered set of indices whose tokens have been inserted. The interpolant $x_t$ reveals the true token $x_1^i$ only after both insertion and unmasking. Given access to this ordered set, one interpolating rate is:

**Proposition D.2** (FlexMDM Interpolating Rate). *A joint interpolating rate matrix $Q_t^{x_0, x_1}$ for the joint interpolant above is given by:*

1. *Unmask: For index set $s$, $x \subseteq x_1|_s$, $x^i = \mathbf{m}$, and $v \in \Sigma$:*

$$Q_t^{x_1}\big((x,s),(x[x^i \leftarrow v],s)\big) = \frac{\dot{\beta}_t}{1-\beta_t} \cdot \mathbf{1}\{x_1^{s[i]} = v\}$$

2. *Insert: For index set $s$, $x \subseteq x_1|_s$, $j \notin s$, and position $i$ such that $s[i-1] < j < s[i]$:*

$$Q_t^{x_1}\big((x,s),(x \triangleleft_i \mathbf{m}, s \cup \{j\})\big) = \frac{\dot{\alpha}_t}{1-\alpha_t}$$

*Proof.* We first write down the $p_t(\cdot, \cdot|x_1)$ the conditioned pmf of $(s,x)$ given $x_0, x_1$. Let $n = \text{len}(x_t)$, then

$$p_t(s, x \mid x_1) = A(t) \prod_{i \in s_t} I_i(t),$$

$$A(t) := (1-\alpha_t)^{n-|s|}$$

$$I_i(t) := \int_0^t \dot{\alpha}_u \left(\frac{1-\beta_t}{1-\beta_u}\mathbf{1}\{x^i = \mathbf{m}\} + \frac{\beta_t - \beta_u}{1-\beta_u}\mathbf{1}\{x^i = x_1^i\}\right) du.$$

Differentiate using the product rule:

$$\partial_t p_t(s, x \mid x_1) = \dot{A}(t) \prod_{i \in s} I_i(t) + A(t) \sum_{j \in s}\left(\dot{I}_j(t) \prod_{i \in s \setminus \{j\}} I_i(t)\right).$$

For $A(t) = (1-\alpha_t)^{n-|s|}$ we have

$$\dot{A}(t) = -(n-|s|)\dot{\alpha}_t(1-\alpha_t)^{n-|s|-1} = A(t)\left(-\frac{(n-|s|)\dot{\alpha}_t}{1-\alpha_t}\right).$$

For each $i \in s$ apply the Leibniz rule to $I_i(t)$:

$$\dot{I}_i(t) = \dot{\alpha}_t \mathbf{1}\{x^i = \mathbf{m}\} + \int_0^t \dot{\alpha}_u\left(\frac{-\dot{\beta}_t}{1-\beta_u}\mathbf{1}\{x^i = \mathbf{m}\} + \frac{\dot{\beta}_t}{1-\beta_u}\mathbf{1}\{x^i = x_1^i\}\right) du$$

$$= \dot{\alpha}_t \mathbf{1}\{x^i = \mathbf{m}\} + \dot{\beta}_t\left(-\mathbf{1}\{x^i = \mathbf{m}\} + \mathbf{1}\{x^i = x_1^i\}\right)\int_0^t \frac{\dot{\alpha}_u}{1-\beta_u} du.$$

Substituting $\dot{A}(t)$ and $\dot{I}_i(t)$ into the product-rule expansion yields

$$\partial_t p_t(s, x \mid x_1)$$
$$= -(n-|s|)\dot{\alpha}_t(1-\alpha_t)^{n-|s|-1}\prod_{i \in s} I_i(t)$$

$$+ (1-\alpha_t)^{n-|s|}\sum_{j \in s}\left[\left(\dot{\alpha}_t\mathbf{1}\{x^j = \mathbf{m}\} + \dot{\beta}_t\left(-\mathbf{1}\{x^j = \mathbf{m}\} + \mathbf{1}\{x^j = x_1^j\}\right)\int_0^t \frac{\dot{\alpha}_u}{1-\beta_u} du\right)\right.$$

$$\left. \cdot \prod_{i \in s \setminus \{j\}} I_i(t)\right]$$

$$= -(n-|s|)\frac{\dot{\alpha}_t}{1-\alpha_t}A(t)\prod_{i \in s} I_i(t) - \sum_{j \in s, x^j = \mathbf{m}}\frac{\dot{\beta}_t}{1-\beta_t}A(t)\prod_{i \in s} I_i(t)$$

$$+ \sum_{j \notin s, x^k \neq \mathbf{m}}\frac{\dot{\beta}_t}{1-\beta_t}A(t)\left(\int_0^t \dot{\alpha}_u \frac{1-\beta_t}{1-\beta_u} du\right)\prod_{i \in s-\{j\}} I_i(t) + \sum_{i \in s}\frac{\dot{\alpha}_t}{1-\alpha_t}(1-\alpha_t)A(t)\prod_{i \in s-\{j\}} I_i(t)$$

This can then be rewritten term by term as,

$$\partial_t p_t(s, x \mid x_1) = -\sum_{i \notin s} \frac{\dot{\alpha}_t}{1 - \alpha_t} p_t(s, x \mid x_1) - \sum_{x^i = \mathbf{m}} \frac{\dot{\beta}_t}{1 - \beta_t} p_t(s, x \mid x_1)$$

$$+ \sum_{x^i \neq \mathbf{m}} \frac{\dot{\beta}_t}{1 - \beta_t} p_t(s, x[x^i \leftarrow \mathbf{m}] | x_1) + \sum_{x^i = \mathbf{m}} \frac{\dot{\alpha}_t}{1 - \alpha_t} p_t(s - \{s[i]\}, \text{remove}(x, i) \mid x_1)$$

where $\text{remove}(x, i)$ refers to the string constructed by removing the $i$-th element of $x$.

Notice that this is equivalent to the R.H.S of the KFE (Eq. 3) if one uses the rate matrix $Q_t^{x_1}$. The four terms correspond to **1)** Outgoing mass from insertion; **2)** Outgoing mass from unmasking; **3)** Incoming mass from unmasking; **4)** Incoming mass from insertion. This concludes the proof. $\square$

**Proposition D.3** (FlexMDM Rate Matrix (Restated from Proposition 2)). *By Proposition D.1, the induced marginal target rate $R_t$ is:*

*1. Unmask: For $x^i = \mathbf{m}$, $v \in \Sigma$:*

$$R_t\left(x, x[x^i \leftarrow v]\right) = \frac{\dot{\beta}_t}{1 - \beta_t} \cdot \mathbb{P}(x_1^{s_t[i]} = v \mid x_t = x)$$

*2. Insert: For position $i \in \{0, \ldots, |x|\}$:*

$$R_t\left(x, x \triangleleft_i \mathbf{m}\right) = \frac{\dot{\alpha}_t}{1 - \alpha_t} \cdot \mathbb{E}_{s_t} \left[s_t[i] - s_t[i - 1] - 1 \mid x_t = x\right]$$

*Proof.* The proof follows by noting proposition D.2 and invoking proposition D.1. $\square$

Performing an approximate target rate matrix $\hat{R}_t$ in terms of an approximate posterior by token $f_\theta(x, t)[i, v] \approx \mathbb{P}(x_1^{s_t[i]} = v \mid x_t = x)$ and an approximate number of insertions $g_\theta(x, t)[i] \approx \mathbb{E}_{s_t} \left[s_t[i] - s_t[i - 1] - 1 \mid x_t = x\right]$. The target rate matrix can be learned by minimizing the following variational objective. Note that the variational loss objective below is the same as one we defined in equation 7.

**Proposition D.4** (FlexMDM Loss (Restated from Proposition 1)). *The loss function is defined as:*

$$L[\hat{R}_t] = \int_0^1 \mathbb{E}_{x_1, s_t, x_t} \left[ -\frac{\dot{\beta}_t}{1 - \beta_t} \sum_{i=0}^{\text{len}(x_t) - 1} \mathbf{1}\{x_t^i = \mathbf{m}\} \log f_\theta(x_t, t)[i, x_1^{s_t[i]}] \right] dt,$$

$$+ \int_0^1 \mathbb{E}_{x_1, s_t, x_t} \left[ \frac{\dot{\alpha}_t}{1 - \alpha_t} \sum_{i=0}^{|x_1|} \phi(s_t[i] - s_t[i - 1] - 1, g_\theta(x_t, t)[i]) \right] dt,$$

*where $\phi(x, y) = y - x + x \log \frac{x}{y}$, is uniquely minimized when $\hat{R}_t = R_t$ and is connected by terminal KL-divergence by:*

$$\mathcal{D}_{KL}(p_1 || \hat{p}_1) \leq L[\hat{R}_t],$$

*where $\hat{p}_1$ is the approximate data distribution induced by $\hat{R}_t$.*

*Proof.* Let $\mathbb{P}$ and $\hat{\mathbb{P}}$ be the path measure of a continuous time Markov chain starting with the empty string with rate matrix $R_t$ and $\hat{R}_t$, respectively.

Consider the KL-divergence between path measures $\mathbb{P}$ and $\hat{\mathbb{P}}$,

$$\mathcal{D}_{\mathrm{KL}}(\mathbb{P} \,||\, \hat{\mathbb{P}}) = \mathbb{E}_{\mathbb{P}} \left[ \int_{t=0}^{t=1} R_t(x_t, x_t) - \hat{R}_t(x_t, x_t) \, dt + \sum_{t: x_t \neq x_{t-}} \log \frac{R_t(x_{t-}, x_t)}{\hat{R}_t(x_{t-}, x_t)} \right]$$

$$= \int_0^{t=1} \mathbb{E}_{x_t} \left[ \sum_{y \neq x_t} \hat{R}_t(x_t, y) - \sum_{y \neq x_t} R_t(x_t, y) - R_t(x_t, y) \log \frac{R_t(x_t, y)}{\hat{R}_t(x_t, y)} \right] dt$$

$$= \int_0^1 \mathbb{E}_{x_1, s_t, x_t} \left[ -\frac{\dot{\beta}_t}{1 - \beta_t} \sum_{i=0}^{\mathrm{len}(x_t)-1} \mathbf{1}\{x_t^i = \mathbf{m}\} \log f_\theta(x_t, t)[i, x_1^{s_t[i]}] \right] dt$$

$$+ \int_0^1 \mathbb{E}_{x_1, s_t, x_t} \left[ \frac{\dot{\alpha}_t}{1 - \alpha_t} \sum_{i=0}^{|x_1|} \phi(s_t[i] - s_t[i-1] - 1, g_\theta(x_t, t)[i]) \right] dt$$

The terminal KL-bound then follows from the data processing inequality, that is:

$$\mathcal{D}_{\mathrm{KL}}(p_1 || \hat{p}_1) \leq \mathcal{D}_{\mathrm{KL}}(\mathbb{P} || \hat{\mathbb{P}})$$

$\square$

# E   DETAILS FOR SECTION 4

## E.1   PRECISE DETAIL ON THE INFERENCE ALGORITHMS

In this section, we provide details on the unmasking steps of vanilla and adaptive inference for FlexMDM, summarized in Algorithm 1, Subroutine 2. Suppose at inference time we are given the discretization step size $\tau$, a partially observed sequence $X_{t_k}$, and the current time step $t_k$.

**Vanilla inference.**   For each masked position $i$ (i.e., $X_{t_k}^i = \mathbf{m}$) and each clean token $v \in \Sigma$, we sample unmasking events from a Poisson distribution $\mathrm{Poisson}(R_v \tau)$, where $R_v$ is the unmasking rate toward token $v$. Concretely, $R_v = \frac{\dot{\beta}_{t_k}}{1 - \beta_{t_k}} \cdot f_\theta(X_{t_k}, t_k)[i, v]$, so that the event count is distributed as $k_v \sim \mathrm{Poi}\left(\tau \cdot \frac{\dot{\beta}_{t_k}}{1 - \beta_{t_k}} \cdot f_\theta(X_{t_k}, t_k)[i, v]\right)$. A masked position is unmasked only if *exactly one* token $v$ produces a count $k_v = 1$ while all others produce zero. This tau-leaping scheme batches all events that occur within the interval $[t_k, t_k + \tau]$.

**Adaptive inference.**   We first draw the number of tokens to unmask, denoted by an integer $K$. While Proposition 3 (Theorem E.1) shows that the choice of $K$ does not affect the theoretical guarantees, in practice, we set $K$ to match the expected number of unmasked tokens under vanilla inference yields stable behavior. Accordingly, we sample $K \sim \mathrm{Poi}\left(\tau \cdot \frac{\dot{\beta}_{t_k}}{1 - \beta_{t_k}} \cdot \#\{\text{masked tokens in } X_{t_k}\}\right)$.

Next, we compute a confidence score for each masked position, based on heuristics such as:

- Top-K probability (Chang et al., 2022; Zheng et al., 2024): For state $x$ at time $t$, the confidence at position $i$ is given by $\max_{v \in \Sigma} f_\theta(x, t)[i, v]$.
- Top-K probability with sliding window: We further restrict sampling to the leftmost $L$ tokens, where
$$L = \min(\lfloor \gamma_1 \cdot L \rfloor, \gamma_2),$$
with $\gamma_1$ and $\gamma_2$ hyperparameters. This approach is related to semi-autoregressive strategies used in Nie et al. (2025).

Finally, we select the subset of positions to unmask as the Top-K masked indices with the highest confidence scores.

### E.2 PROOF OF FLEXMDM'S ANY-ORDER INFERENCE CAPABILITY

#### E.2.1 PROOF PRELIMINARIES

**Form of posterior.** We first compute, for each $x_t^i$, the probabilities of being masked or deleted in equation (6). This follows from a straightforward calculation using the joint distribution of $(T_1^i, T_2^i)$:

$$p(x_t^i = (\text{empty})) = p(T_1^i > t) = 1 - \alpha_t,$$

$$p(x_t^i = \mathbf{m}) = p(T_1^i \le t, T_2^i > t) = \int_t^1 \int_0^t \left( \frac{\dot{\beta}_s}{1 - \beta_u} \times \dot{\alpha}_s \right) ds\, du =: 1 - \gamma_t.$$

Here we define $\gamma_t$ as $1 - p(x_t^i = \mathbf{m})$. Therefore, the process in equation (6) is equivalent to observing a partially masked subsequence $x_t$ obtained by sampling $x_1 \sim p$ and, for each position of $x_1$, independently deleting it with probability $1 - \alpha_t$, masking it with probability $1 - \gamma_t$, or leaving it unchanged with probability $\alpha_t + \gamma_t - 1$. Note that $\alpha_t$ and $\gamma_t$ both increase from 0 to 1 as $t$ increases from 0 to 1.

The posterior is given by

$$p(x_1 = x^* \mid x_t = x)$$
$$\propto p(x^*) \cdot p(x_t = x \mid x_1 = x^*)$$
$$= p(x^*) \cdot (1 - \alpha_t)^{\text{len}(x^*) - \text{len}(x)} (1 - \gamma_t)^{\#\text{mask}(x)} (\alpha_t + \gamma_t - 1)^{\#\text{unmask}(x)} \cdot \#\{s : x \subseteq x^*|_s\}$$
$$\propto p(x^*) \cdot (1 - \alpha_t)^{\text{len}(x^*) - \text{len}(x)} \cdot \#\{s : x \subseteq x^*|_s\}.$$

Importantly, as in the vanilla MDM setting, the posterior does not depend on the unmasking schedule $(\gamma_t)$ (thus $\beta_t$), which will enable us to perform unmasking in adaptively chosen positions. Note also that if all sequences in the support of $p$ were of the same length, this posterior would also be independent of $(\alpha_t)$; while we do not prove it, in this case this would allow us to choose an arbitrary order of unmaskings and insertions.

**Extension of posterior to $t = 1$.** Motivated by the form of the posterior above, we define the following:

$$q_t(x^* \mid x) \propto \begin{cases} p(x^*) \cdot \mathbf{1}_{x \subseteq x^*} & \text{if } t = 1 \\ p(x^*) \cdot (1 - \alpha_t)^{\text{len}(x^*) - \text{len}(x)} \cdot \#\{s : x \subseteq x^*|_s\} & \text{otherwise} \end{cases}$$

Note that for $t < 1$, this is the same as $p(x_1 = x \mid x_t = x)$. We will denote the marginals of $q_t(\cdot \mid x)$ by $q_t^i(\cdot \mid x)$ for $v \in \Sigma$. The reason for extending the definition of the posterior to $t = 1$ is that in an adaptive FlexMDM sampler (see Definition E.1), because we are entirely decoupling unmasking from the schedule of insertions, after the final insertion step the time parameter $t$ may be 1 even though there are still tokens left to unmask. We will assume oracle access to $q_1(\cdot \mid x)$ as in practice these are simply the any-order marginals for $p$, and furthermore in practice these are already well-approximated by the learned posterior marginals $p(x_1^i \cdot \mid x_{1-\delta} = x)$ for arbitrarily small $\delta > 0$.

**Index-tracking variable.** Recall that in the definition of the joint interpolant we defined an index-tracking variable $s_t$ which essentially tracked which indices of $x_1$ correspond to the tokens in $x_t$. While our analysis below will not use the language of stochastic interpolants, we will still use the idea of tracking $s_t$, with slightly modified notation. Specifically, for any $0 \le t < 1$, we will use the notation $\Pr_{(x_1,s)|x_t=x}$ and $\mathbb{E}_{(x_1,s)|x_t=x}$ to denote probability and expectation with respect to the distribution given by sampling $x_1$ from $q_t(\cdot \mid x)$, and then sampling $s$ uniformly random from subsets for which $x \subseteq (x_1)|_s$. When we only care about the marginal distribution over $s$, we write $\Pr_{s|x_t=x}$ and $\mathbb{E}_{s|x_t=x}$. Given such a subset $s$ and $i \in \{0, \ldots, |s| - 1\}$, we use $s_i$ to denote its $i$-th element in sorted order; as before, we also define the boundary values $s_{-1} = -1$ and $s_{\text{len}(s)} = \text{len}(x_1)$. The insertion expectations which we had denoted by $\mathbb{E}[s_t[i] - s_t[i - 1] - 1]$ in the main body are thus given by $\mathbb{E}_{s|x_t=x}[s_i - s_{i-1} - 1]$ in the notation of this section.

### E.3 FORMAL GUARANTEE FOR ADAPTIVE INFERENCE

**Definition E.1.** *Given query access to the posterior marginals $q_t^i(\cdot \mid x_t = x)$ and to the insertion expectations $\mathbb{E}_{s|x_t=x}[s_i - s_{i-1} - 1]$, an* adaptive FlexMDM sampler *is any algorithm which produces*

a sequence of iterates $\hat{x}_{t_1}, \ldots, \hat{x}_{t_n}$, where $0 = t_1 < \cdots < t_n = 1$, by starting at $\hat{x}_{t_1} = \varepsilon$ and arbitrarily alternating between steps of the following form:

- Any-order *unmasking step: Starting from $\hat{x}_{t_j}$, if $\mathrm{mask}(\hat{x}_{t_j})$ is nonempty, pick an arbitrary index $i$ therein (possibly probabilistically), sample $v$ from $q_{t_j}^i(\cdot \mid \hat{x}_{t_j})$, and set $\hat{x}_{t_j} \leftarrow \hat{x}_{t_j}[\hat{x}_{t_j}^i \leftarrow v]$.*

- *Insertion step: Starting from $\hat{x}_{t_j}$, run the CTMC with rate matrix*

$$R_t^{\mathrm{ins}}(x, y) = \begin{cases} \mathbb{E}_{s|x_t=x}[s_i - s_{i-1} - 1] \cdot \frac{\dot{\alpha}_t}{1 - \alpha_t} & \text{if } y = x \triangleleft_i \mathbf{m} \\ -\sum_{i=0}^{\mathrm{len}(x)} R_t^{\mathrm{ins}}(x, x \triangleleft_i \mathbf{m}) & \text{if } y = x \\ 0 & \text{otherwise} \end{cases} \tag{9}$$

for $t_j \le t \le t_{j+1}$ to obtain $\hat{x}_{t_{j+1}}$. If $t_{j+1} = 1$, then apply any-order unmasking until $\mathrm{mask}(\hat{x}_{t_{j+1}})$ is empty, and terminate.

Note that the rate matrix in the second bullet point above is identical to the one in the main body except that we only consider transitions given by insertions.

Formally, we will show the following:

**Theorem E.1.** *Any adaptive FlexMDM sampler for $p$ will generate a sequence of iterates $\hat{x}_{t_1}, \ldots, \hat{x}_{t_n}$ such that $\hat{x}_{t_n}$ is exactly a sample from $p$.*

### E.4 PROOF OF THEOREM E.1

To show that adaptive sampling works, we inductively prove an even stronger statement: at any intermediate step of the sampler after it has produced $\hat{x}_{t_j}$, the final output $\hat{x}_{t_n}$ is a sample from $q_{t_j}(\cdot \mid \hat{x}_{t_j})$.

The following two lemmas provide the inductive steps for unmasking and insertion respectively:

**Lemma E.1** (Inductive step for unmasking). *Let $0 \le t \le 1$ and let $x$ be a partially unmasked subsequence of length $m$. Let $\pi = (\pi_i)_{i \in \mathrm{mask}(x)}$ denote any distribution over masked indices of $x$. Suppose that one runs the following:*

1. *Sample index $i$ from $\pi$*

2. *Sample $v$ from the posterior marginal $q_t^i(\cdot \mid x_t = x)$*

3. *Sample from $q_t(\cdot \mid x[x^i \leftarrow v])$.*

*The output of this procedure is a sample $x^*$ from $q_t(\cdot \mid x)$.*

**Lemma E.2** (Inductive step for insertion). *Let $0 \le t < 1$ and let $x$ be a partially unmasked subsequence of length $m$. Let $0 \le h \le 1 - t$ be any duration of time. Suppose that one runs the following:*

1. *Starting from $x$, run the CTMC with rate matrix given by Eq. (9) for time $h$ to obtain $x'$*

2. *Sample from $q_{t+h}(\cdot \mid x')$.*

*The output of this procedure is a sample from the posterior $q_t(\cdot \mid x)$.*

We defer the proofs of these to Sections E.5 and E.6 below. The idea for the former is identical to the proof of the folklore fact that vanilla MDMs can sample in any order (Kim et al., 2025). The proof for the latter is more involved and involves explicitly verifying that the Kolmogorov *backward* equation is satisfied by the rate matrix we have constructed.

Here we verify that these Lemmas are enough to establish Theorem E.1.

*Proof of Theorem E.1.* We show more generally that starting from any intermediate time step $\hat{x}_{t_j}$ (not just $j = 1$), any adaptive FlexMDM sampler outputs a sample from $q_{t_j}(\cdot \mid \hat{x}_{t_j})$. We do this by inducting on the total number of insertion steps that remain.

As the base case for the induction, if no more insertion steps remain, then we must have $t_j = 1$. In this case, we can further induct on the number of unmasking steps and apply Lemma E.1 with $t$ therein set to 1 to conclude that the final output is a sample from $q_1(\cdot \mid \hat{x}_{t_j})$.

For the inductive step, we have $t_j < 1$ and suppose we have shown that for any FlexMDM sampler that makes at most $m$ insertion steps, starting from any $\hat{x}_{t_j}$ at intermediate time $t_j$, it samples from $q_{t_j}(\cdot \mid \hat{x}_{t_j})$. Now consider a FlexMDM sampler that makes at most $m + 1$ insertion steps starting from $\hat{x}_{t_j}$ at intermediate time $t_j$. If in the next step it performs an insertion step, i.e. it runs the CTMC with rate matrix defined above for total time $h = t_{j+1} - t_j$, then by Lemma E.2 and the inductive hypothesis, it samples from $q_{t_j}(\cdot \mid \hat{x}_{t_j})$. Alternatively, suppose the sampler performs some sequence of $\ell$ unmasking steps before performing an insertion step. Then by further inducting on $\ell$, we conclude by Lemma E.1 that the sampler eventually outputs a sample from $q_{t_j}(\cdot \mid \hat{x}_{t_j})$.

Finally, the theorem follows from the special case where $t_j = 0$ and $\hat{x}_{t_j} = \varepsilon$. $\qquad\square$

### E.5 PROOF OF LEMMA E.1

*Proof.* Fix any index $i \in \text{mask}(x)$. The marginal $q_t^i(\cdot \mid x)$ is given by

$$\Pr_{(x_1, S) \mid x_t = x} [(x_1)|_{s_i} = v] = \frac{\sum_{x_1, S:(x_1)|_{s_i} = v} p(x) \cdot (1 - \alpha_t)^{\text{len}(x_1) - \text{len}(x)} \cdot \#\{S : x \subseteq (x_1)|_S\}}{\sum_{x_1} p(x_1)(1 - \alpha_t)^{\text{len}(x_1) - \text{len}(x)} \cdot \#\{S : x \subseteq (x_1)|_S\}}. \tag{10}$$

The posterior $q_t(\cdot \mid x[x^i \leftarrow v])$ is given by

$$q_t(x^* \mid x[x^i \leftarrow v]) = \frac{p(x^*) \cdot (1 - \alpha_t)^{\text{len}(x^*) - \text{len}(x)} \cdot \#\{S : x[x^i \leftarrow v] \subseteq x^*|_S\}}{\sum_{x_1} p(x_1) \cdot (1 - \alpha_t)^{\text{len}(x_1) - \text{len}(x)} \cdot \#\{S : x[x^i \leftarrow v] \subseteq (x_1)|_S\}}. \tag{11}$$

Note that the numerator of Eq. (10) and the denominator of Eq. (11) are the same. So conditioned on unmasking index $i$, the above procedure outputs $x^*$ with probability

$$\sum_z \Pr_{(x_1, S) \mid x_t = x} [(x_1)|_{s_i} = v] \cdot q_t(x^* \mid x[x^i \leftarrow v])$$
$$= \frac{p(x^*) \cdot (1 - \alpha_t)^{\text{len}(x^*) - \text{len}(x)} \cdot \sum_z \#\{S : x[x^i \leftarrow v] \subseteq x^*|_S\}}{\sum_{x_1} p(x_1)(1 - \alpha_t)^{\text{len}(x_1) - \text{len}(x)} \cdot \#\{S : x \subseteq (x_1)|_S\}}$$
$$= q_t(x^* \mid x).$$

This holds conditioned on unmasking any $i \in \text{mask}(x)$, so regardless of the choice of $\pi$ over such positions, we will sample from the correct distribution $q_t(\cdot \mid x)$. $\qquad\square$

### E.6 PROOF OF LEMMA E.2

*Proof.* It suffices to show that the rate matrix satisfies the Kolmogorov *backward* equation

$$\partial_t q_t(x^* \mid x) = -\sum_{i=0}^{\text{len}(x)} R_t^{\text{ins}}(x, x \lhd_i \mathbf{m}) q_t(x^* \mid x \lhd_i \mathbf{m}) - R_t^{\text{ins}}(x, x) q_t(x^* \mid x).$$

First note that the rate $R_t^{\text{ins}}(x, x \lhd_i \mathbf{m})$ can be expressed as

$$\frac{\sum_{x_1} p(x_1) \cdot (1 - \alpha_t)^{\text{len}(x_1) - \text{len}(x)} \cdot \sum_{S:x \subseteq (x_1)|_S} (s_i - s_{i-1} - 1)}{\sum_{x_1} p(x_1) \cdot (1 - \alpha_t)^{\text{len}(x_1) - \text{len}(x)} \cdot \#\{S : x \subseteq (x_1)|_S\}} \cdot \frac{\dot{\alpha}_t}{1 - \alpha_t}.$$

Furthermore, $\sum_i (s_i - s_{i-1} - 1) = \text{len}(x_1) - \text{len}(x)$, so

$$\sum_i R_t^{\text{ins}}(x, x \lhd_i \mathbf{m})$$
$$= \frac{\sum_{x_1} p(x_1) \cdot (1 - \alpha_t)^{\text{len}(x_1) - \text{len}(x)} \cdot \#\{S : x \subseteq (x_1)|_S\} \cdot (\text{len}(x_1) - \text{len}(x))}{\sum_{x_1} p(x_1) \cdot (1 - \alpha_t)^{\text{len}(x_1) - \text{len}(x)} \cdot \#\{S : x \subseteq (x_1)|_S\}} \cdot \frac{\dot{\alpha}_t}{1 - \alpha_t}. \tag{12}$$

Let us compute $\partial_t q_t(x^* \mid x)$:

$$
\begin{aligned}
&- \frac{p(x^*) \cdot (1 - \alpha_t)^{\mathrm{len}(x^*) - \mathrm{len}(x)} \cdot \#\{S : x \subseteq x^*|_S\} \cdot (\mathrm{len}(x^*) - \mathrm{len}(x))}{\sum_{x_1} p(x_1) \cdot (1 - \alpha_t)^{\mathrm{len}(x_1) - \mathrm{len}(x)} \cdot \#\{S : x \subseteq (x_1)|_S\}} \cdot \frac{\dot{\alpha}_t}{1 - \alpha_t} \\
&+ \Bigg[ \frac{\sum_{x_1} p(x_1) \cdot (1 - \alpha_t)^{\mathrm{len}(x_1) - \mathrm{len}(x)} \cdot \#\{S : x \subseteq (x_1)|_S\} \cdot (\mathrm{len}(x_1) - \mathrm{len}(x))}{\sum_{x_1} p(x_1) \cdot (1 - \alpha_t)^{\mathrm{len}(x_1) - \mathrm{len}(x)} \cdot \#\{S : x \subseteq (x_1)|_S\}} \cdot \frac{\dot{\alpha}_t}{1 - \alpha_t} \\
&\qquad \times \frac{p(x^*) \cdot (1 - \alpha_t)^{\mathrm{len}(x^*) - \mathrm{len}(x)} \cdot \#\{S : x \subseteq x^*|_S\}}{\sum_{x_1} p(x_1) \cdot (1 - \alpha_t)^{\mathrm{len}(x_1) - \mathrm{len}(x)} \cdot \#\{S : x \subseteq (x_1)|_S\}} \Bigg]
\end{aligned} \tag{13}
$$

Note that by Eq. (12), the term in the parentheses in Eq. (13) is exactly

$$
\sum_{i=0}^{\mathrm{len}(x)} R_t^{\mathrm{ins}}(x, x \triangleleft_i \mathbf{m}) q_t(x^* \mid x) = -R_t^{\mathrm{ins}}(x, x) q_t(x^* \mid x),
$$

It remains to verify that the first term in Eq. (13) is equal to $-\sum_i R_t^{\mathrm{ins}}(x, x \triangleleft_i \mathbf{m}) q_t(x^* \mid x \triangleleft_i \mathbf{m})$. To that end, we must show that

$$
\sum_{i=0}^{\mathrm{len}(x)} \frac{\sum_{x_1} p(x_1)(1 - \alpha_t)^{\mathrm{len}(x_1) - \mathrm{len}(x)} \cdot \sum_{S : x \subseteq (x_1)|_S} (s_i - s_{i-1} - 1)}{\sum_{x_1} p(x_1)(1 - \alpha_t)^{\mathrm{len}(x_1) - \mathrm{len}(x)} \cdot \#\{S : x_t \triangleleft_i \mathbf{m} \subseteq (x_1)|_S\}} \cdot \#\{S : x \triangleleft_i \mathbf{m} \subseteq x^*|_S\}
$$
$$
= \#\{S : x \subseteq (x^*)|_S\} \cdot (\mathrm{len}(x^*) - \mathrm{len}(x)) \tag{14}
$$

The key combinatorial step is as follows. For any $x_1$ in the support of $p$, consider a subset $S$ for which $x \subseteq (x_1)|_S$. Note that for every such $S$, we can uniquely associate exactly $s_i - s_{i-1} - 1$ different subsets $S'$ of size $|S| + 1$ for which $x \triangleleft_i \mathbf{m} \subseteq (x_1)|_{S'}$. Therefore, $\sum_{S : x \subseteq (x_1)|_S} (s_i - s_{i-1} - 1) = \#\{S : x \triangleleft_i \mathbf{m} \subseteq (x_1)|_S\}$, and the left-hand side of Eq. (14) thus becomes

$$
\sum_{i=0}^{\mathrm{len}(x)} \#\{S : x \triangleleft_i \mathbf{m} \subseteq x^*|_S\} = \sum_{i=0}^{\mathrm{len}(x)} \sum_{S : x \subseteq x^*|_S} (s_i - s_{i-1} - 1) = \sum_{S : x \subseteq (x_1)|_S} (\mathrm{len}(x^*) - \mathrm{len}(x_t)),
$$

which completes the proof of Eq. (14). $\qquad\square$

## F EXPERIMENTAL DETAILS

### F.1 PRETRAINING ON OPENWEBTEXT

**Dataset preparation.** As mentioned in Section 5.1, to obtain a variable-length dataset, we split OpenWebText articles paragraph-wise using the GPT-2 tokenizer Radford & Wu (2019). This can be implemented by locating the token index corresponding to the delimiter \n \n. Sequences longer than 1024 tokens are then chunked recursively by splitting by the delimiter closest to the middle sequence, yielding a variable-length dataset with maximum sequence length 1024.

**FlexAttention.** To handle variable-length sequences during training, we pad each batch to the maximum sequence length. In MDM, by design, pad tokens also enter the model input. In contrast, FlexMDM is designed to receive only clean or mask tokens as inputs. Ideally, QKV attention should not attend to pad tokens; however, current FlashAttention (Dao et al., 2022) does not support this for *non-causal* attention (our setup of interest). We therefore adapt FlexAttention (Dong et al., 2024). A side benefit is improved training speed, since FlexMDM performs attention over fewer tokens than MDM's full-sequence attention. Note that in the LLaDA experiment, we did not implement this optimization; pad tokens can therefore attend to other tokens, though we expect the impact to be negligible.

**Training configuration.** As in Section 5.1, we model FlexMDM with a DiT Peebles & Xie (2023) backbone and embed the insertion schedule $\alpha_t$. For MDM, we use the same DiT configuration with time step embedding but without the softplus scalar head. Transformer configuration is: `hidden size:768`, `heads:12`, `conditional dimension:128`, `dropout:0.05`, `layers:12`.

We train both models on 16 H100 GPUs with a global batch size of 1024 and max training iteration $1M$. We use the AdamW (Loshchilov & Hutter, 2017) optimizer with weight decay 0.03, learning rate 3e$-$4, 2000 warmup steps, and an EMA factor of 0.9999. Additionally, we use low-discrepancy time-step sampling to reduce variance: one $t$ is drawn uniformly from each interval $[i/T, (i+1)/T]_{i=0}^{T-1}$, as in prior MDM training (Shi et al., 2024).

**Metric.** For evaluation, we take sequences generated by both models and retain the clean tokens by removing padding (e.g., the leading pad token). We adopt LLaMA-2-7B (Touvron et al., 2023) as the reference model to compute likelihoods. We notice that the pretrained MDM generates short sentences with unreasonably large (worse) perplexities. Therefore, we filter overly short sequences with $\leq 10$ tokens when calculating average perplexity.

### F.1.1 ADDITIONAL EXPERIMENTS

**Sequence entropy.** We follow the setup from (Zheng et al., 2024) to measure the entropy of given sequence. As we can observe in the table below, both models exhibit comparable sequence entropy across different sampling steps.

| Sampling steps | 256 | 512 | 1024 | 2048 | 4096 |
|---|---|---|---|---|---|
| MDM | 4.4732 | 4.5691 | 4.4417 | 4.2259 | 4.1876 |
| FlexMDM | 4.4586 | 4.5173 | 4.5660 | 4.4712 | 4.4304 |

**Generative perplexity for extended sampling steps.** Below, we extend the generative perplexity measurement to more sampling steps. We observe plateaus around 16,384 steps. This is likely because discretization errors for the CTMC and the estimation errors for each transition entry become negligible, leading to diminishing generative perplexity.

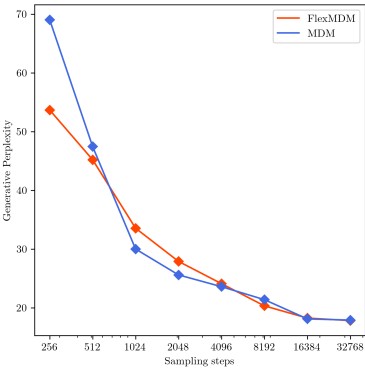

**Downstream evaluation on the text understanding.** We follow the evaluation protocol of von Rütte et al. (2025).

| Model | HellaSwag | PIQA | ARC-E | ARC-C | BoolQ | OBQA | WinoGrade |
|---|---|---|---|---|---|---|---|
| FlexMDM | 28.75 | 51.69 | 25.26 | 25.75 | **50.50** | **24.60** | **51.78** |
| MDM | **29.70** | **52.55** | 25.26 | **26.09** | 50.05 | 23.80 | 49.01 |

### F.2 PRETRAINING ON THE MAZE PLANNING TASK

**Task construction.** We generate mazes with a fully random, recursive division procedure (the code is provided in Listing 1), on a $41 \times 41$ grid, with some invalid cells. As described in Section 5.1, we consider a subgoal-conditioned planning task: the model is given a sequence of subgoals $(g_1, \ldots, g_K)$ and must produce a valid path that connects them in order. To construct training examples for a given $K$, we sample 15000 start–goal pairs $(g_1, g_K)$, compute the shortest path for each pair via breadth-first search, and then add controlled detours to obtain up to 10 distinct valid paths per pair. Subgoals are formed by selecting $K-2$ intermediate cells uniformly at random along

a chosen path (start and goal are already fixed). We use $95\%$ of the pairs for training and hold out $5\%$ for validation to evaluate generalization to unseen pairs and subgoal sets.

**Training data construction.** For MDM, the training sequence is $((g_1, \ldots, g_K)$ [SEP] Path$)$, where [SEP] is a special separator and Path denotes the tokenized path. During training, the prompt $(g_1, \ldots, g_K)$ bypasses the interpolant so that, at inference time, the model can condition on $(g_1, \ldots, g_K)$ [SEP] and generate the path. For FlexMDM, we use an interpolant in which the subgoal tokens are exempt from the process in (6); that is, tokens corresponding to each $g_i$ are kept clean at all times. This also enables generation to start from $(g_1, \ldots, g_K)$. Although this conditional generation template changes the base distributions $p_0$ for both MDM and FlexMDM, we note that the theoretical guarantee from Section 3 and Section 4 still holds–once the training is perfect (under the access to the ground truth unmasking posterior and insertion expectation), the inference algorithms recover the ground truth distribution $p_1$.

**Training configuration.** We use the same architectural design as in the OpenWebText pretraining, but with a smaller model: `hidden size:256, heads:8, conditional dimension:128, dropout:0.1, layers:8`. Both models have approximately $8.5M$ parameters. We train them on $4$ A100 GPUs with a global batch size of $256$ for up to $100$ epochs. We use AdamW (Loshchilov & Hutter, 2017) with weight decay $0.01$, learning rate $1 \times 10^{-4}$, $20$ warmup epochs, and an EMA factor of $0.9999$.

**Metric.** Given the final conditionally generated sequence, we report the success rate under two criteria: (1) all visited cells are valid (no collisions with invalid cells), and (2) the path connects the subgoals consecutively in order. We perform inference both models with the number of sampling steps $256$.

Listing 1: Code for the maze Construction

```python
# ----------------------------------------------------------------------
#  RECURSIVE DIVISION (perfect maze) -----------------------------------
# ----------------------------------------------------------------------
def _divide(g, top, left, h, w):
    if h <= 2 or w <= 2:
        return
    horizontal = w < h  # split the longer dimension
    if horizontal:
        y = random.randrange(top + 1, top + h - 1, 2)
        gap = random.randrange(left, left + w, 2)
        g[y, left:left + w] = 1
        g[y, gap] = 0
        _divide(g, top, left, y - top, w)
        _divide(g, y + 1, left, top + h - y - 1, w)
    else:
        x = random.randrange(left + 1, left + w - 1, 2)
        gap = random.randrange(top, top + h, 2)
        g[top:top + h, x] = 1
        g[gap, x] = 0
        _divide(g, top, left, h, x - left)
        _divide(g, top, x + 1, h, left + w - x - 1)

# ----------------------------------------------------------------------
#  WRAPPER with extra doorways ----------------------
# ----------------------------------------------------------------------
def division_maze(m, n, seed=2025, extra_door_frac=0.5):
    """
    m, n              # size in *cells*     (not bitmap coords)
    seed              # int or None
    extra_door_frac  # 0, perfect maze; >0 flicks more doors (loops)
    """
    random.seed(seed)
    H, W = 2 * m + 1, 2 * n + 1
```

```
g = np.zeros((H, W), dtype=np.uint8)
g[0, :], g[H - 1, :], g[:, 0], g[:, W - 1] = 1, 1, 1, 1

_divide(g, 1, 1, H - 2, W - 2)

# ---------- optional imperfection: punch extra doorways ------------
if extra_door_frac > 0:
    candidates = []
    for r in range(1, H - 1):
        for c in range(1, W - 1):
            if g[r, c] == 1:
                # Check if wall separates two passages orthogonally
                if g[r - 1, c] == 0 and g[r + 1, c] == 0:
                    candidates.append((r, c))
                elif g[r, c - 1] == 0 and g[r, c + 1] == 0:
                    candidates.append((r, c))
    k = int(len(candidates) * extra_door_frac)
    for (r, c) in random.sample(candidates, k=k):
        g[r, c] = 0
return g
```

### F.3 WEIGHT INITIALIZATION TRAINING FROM LLADA

In this section, we describe the procedure for adapting the pretrained LLaDA-8B base model into the FlexMDM.

**Training configuration.** LLaDA uses a bidirectional Transformer without an additional time embedding, leveraging the fact that MDM does not require an explicit time signal (Zheng et al., 2024). For FlexMDM, to model the insertion expectation, we inject a time-embedding pathway via AdaLN (Peebles & Xie, 2023). For parameter efficiency, we tie (share) the four AdaLN parameter sets across the intermediate Transformer layers. We also attach a softplus scalar head to parameterize the insertion expectation.

Next, we attach LoRA adapters (Hu et al., 2022) to every attention projection (q_proj, k_proj, v_proj) and to the unmasking-posterior head. We include LoRA on the unmasking posterior head because the unmasking posteriors differ between MDM and FlexMDM: in FlexMDM, unmasking must account for token shifts induced by insertions. This fine-tuning of the last head is crucial for enabling variable-length generation.

The LoRA configuration that we use is $r = 128$, $\alpha = 128$, and dropout $0.1$. Training runs for $200{,}000$ gradient steps with a batch size of $64$ on 16 H100s, which took approximately 3 days. We optimize with AdamW (Loshchilov & Hutter, 2017) at learning rate $1 \times 10^{-4}$ and weight decay $0.1$, using a cosine warm-restarts scheduler.

**Evaluation on GSM8K.** We instruction-fine-tune (IFT) the FlexMDM base model on the GSM8K training split. To preserve the instruction–answer format, we modify the interpolant in Eq. 6 so that tokens corresponding to the instruction are excluded from the interpolant—these tokens remain fixed for all time steps. We apply the same strategy to obtain the baseline (that is, IFT from LLaDA-base), modifying the MDM interpolant so that instruction tokens remain fixed. This choice is equivalent to the IFT recipe used in Nie et al. (2025); Ye et al. (2025). Both models are IFT-ed for $1000$ epochs. (Other IFT hyperparameters match those used in our base setup unless otherwise noted.)

For FlexMDM inference, we start from the instruction prompt at $t = 0$ and run adaptive inference to $t = 1$. Concretely, we use Top-K probability with a sliding window (Appendix E) with $\gamma_1 = 5.0$ and $\gamma_2 = 64$. For LLaDA, we report the best result under the semi-autoregressive, confidence-based sampling of (Nie et al., 2025). For both models, we set the token sampling temperature to $0.0$, which we confirm to be important for strong Pass@1. Overall, adaptive inference substantially improves performance over vanilla inference.

For a fair comparison, since FlexMDM is not IFT-ed, we IFT LLaDA-Base, rather LLaDA-instruct, this differs from Zhao et al. (2025). We employ zero-shot evaluation, which also differs from Nie et al. (2024).

**Evaluation on HumanEval-infill.** Code infilling conditions on a prefix and suffix, and asks the model to complete the middle part of the code. For FlexMDM, we format training examples as (`Prefix`;`[SEP]`;`[Middle]`;`[SEP]`;`Suffix`), where `[SEP]` is a separator token and `Instruction` describes the infilling task for a model. We modify Eq. (6) so that tokens for `Prefix`, `Suffix`, and `[SEP]` are fixed in the interpolant (i.e., excluded from the interpolant). Thus, at $t = 0$ the state is (`Prefix`;`[SEP]`;`[SEP]`;`Suffix`).

For MDM, we use the format (`Instruction`;`[PRE]`;`Prefix`;`[SUF ]`;`Suffix`;`[SEP]`;`Middle`), with `Instruction` prompting infill after prefix and suffix, along with `[PRE]` and `[SUF]` separate the prefix and suffix, respectively. Here too, the tokens without `Middle` are held fixed by the modified interpolant. This difference in formatting reflects the fixed-length nature of MDMf MDM (no token insertion). This formatting has been used in the code infilling tasks for autoregressive models in Bavarian et al. (2022). Naively masking the `Middle` span yields a base state at $t = 0$ of (`Prefix`;`Masked`;`Suffix`), where `Masked` is a fully masked sequence of length |`Middle`|. This leaks length information—materially simplifying the task— and renders comparisons to FlexMDM unfair, since FlexMDM does not observe the target span length. For fair evaluation, we therefore avoid length-revealing masks and require methods to infer the span length during inference.

We IFT both models on the educational-instruct split of `opc-sft-stage2`; the architecture and optimization configurations match those used for GSM8K IFT. At evaluation, we initialize from the base distributions: for FlexMDM, (`Instruction`;`Prefix`;`[SEP]`;`[SEP]`;`Suffix`); for MDM, (`Instruction`;`Prefix`;`Suffix`;`[SEP]`). We use the same Top-K adaptive inference for both and temperature $0.0$. Final outputs are scored with the HumanEval-infill verifier toolkit to compute success rates.

