# OpenReview forum: "Any-Order Flexible Length Masked Diffusion"
_ICLR.cc/2026/Conference — ICLR 2026 Poster_

### Official Review · Reviewer_TnF2 · 2025-10-20

**Soundness:** 4
**Presentation:** 4
**Contribution:** 3
**Rating:** 6
**Confidence:** 4

**Summary:**

This paper proposes FlexMDM, a masked diffusion model that jointly models the unmasking posterior and the insertion expectation. Unlike traditional Masked Diffusion Models (MDMs), which begin with a fixed number of masked tokens and progressively unmask them, FlexMDM introduces an additional insertion operation that dynamically inserts new mask tokens during the diffusion process. This design enables variable-length generation while preserving any-order generation capabilities.

FlexMDM can be adapted from an existing MDM with minimal changes—specifically, by adding a scalar output head at each position to predict the insertion expectation.

The authors validate their method across multiple language and planning benchmarks, demonstrating that FlexMDM:

1. Effectively learns the true length distribution of the training data,
2. Achieves comparable generative perplexity to standard MDMs,
3. Outperforms MDMs on planning and reasoning tasks, and
4. Can be efficiently scaled to larger models.

**Strengths:**

1. Writing is very clear and easy to follow
2. The framework is mathematically sound and theoretically grounded and solid
3. The idea is novel and the connection to traditional MDMs are very clear

**Weaknesses:**

I find the experiments relatively weak compared to the strong theoretical foundations of the paper, and several aspects require clarification:

Model comparison: It is unclear what specific model is referred to as “MDM” in the comparisons. For example, is it equivalent to D3PM or another variant?

Figure 4c: The figure (which should include an overall label) shows an interesting trend in perplexity as the number of sampling steps increases. Since neither curve appears to converge by 4096 steps, it would be informative to see what happens beyond this point.

Maze task: While it is reasonable that FlexMDM outperforms traditional MDMs in the maze task, this setup alone seems insufficient to substantiate the claim that it “supports FlexMDM as a principled approach for subgoal-based planning.”

Scaling up experiment: The statement “Surprisingly, we observe rapid transfer: within three days on 16 H100 GPUs, the model generates variable-length sentences” requires clarification. MDMs can already generate variable-length sequences to some extent using padding tokens, as the authors note. It would be useful to clarify in what sense this transfer is “rapid.” Also, while it's an 8B model, the authors only trains a 400M LoRA adapter and claim FlexMDM is "scalable". Can the authors clarify on this?

Ablation study: Including an ablation study on modeling the insertion process would strengthen the empirical evidence. Alternatively, can the authors confirm if the comparison between MDM and FlexMDM already serves this purpose.

**Questions:**

Please see weakness.

---

> ### Author Response · Authors · 2025-11-24
>
> We appreciate the reviewers for their positive evaluations and comments! Below, we address them point by point.
>
> 1. **Model comparison/Study in insertion**: All the MDMs mentioned in the manuscript are the Masked Diffusion Model backbone used in seminal works [Sahoo et al, Shi et al, Nie et al]. As it was demonstrated in multiple works that MDM is the most scalable and efficient approach across different discrete diffusion frameworks, we select it as our preliminary baseline and also a backbone to build FlexMDM on.
>
> *Shi et al. "Simplified and generalized masked diffusion for discrete data, 2024
>
> *Sahoo et al. "Simple and effective masked diffusion language models, 2024
>
> *Nie et al, Large Language Diffusion Models, 2025*.
>
>
> 2. **Additional experiments in perplexity**: We thank the reviewer for the insightful suggestions. As an extended study of sampling steps, we measured generative perplexity for both pretrained models using sampling steps in [8192,16384,32768]. (we attach the figure link below) As expected, perplexity decreases with more sampling steps; however, both models reach a plateau around 16,384 steps. This is intuitive: beyond a certain point, the discretization error for the CTMC and the estimation error for each transition entry become negligible, leading to diminishing returns.
>
> Figure link: https://ibb.co/1t0P2LWC
>
>
> 3. **Scaling-up experiments**: First of all, MDM and FlexMDM can both generate variable-length sequences, but in different ways. We clarify:
> - MDM relies on explicit [EOS] tokens, so although output length can vary, it is still governed indirectly by the maximum sequence length.
> - FlexMDM enables variable-length generation via zero-symbol insertions, providing a strictly more expressive mechanism.
>
> Thus, the pretrained MDM does not natively support true variable-length generation in the same way FlexMDM does. Our claim is that fine-tuning enables the pretrained MDM to rapidly learn insertion behavior, effectively retrofitting it into a FlexMDM. In terms of compute: As reported in Appendix F.3, fine-tuning LLaDA-8B → FlexMDM-8B uses 0.2M gradient steps × batch size 64 × seq length 1024 ≈ 13.1B tokens. This is far smaller than LLaDA’s pretraining compute of 2.3T tokens [Nie et al., 2025]. This supports our claim of rapid transfer: only ~0.6% of the original pretraining tokens are required to retrofit MDM into FlexMDM. We were unable to pretrain a FlexMDM-8B from scratch due to compute limitations; therefore, retrofitting is the most realistic scaling-up experiment we could have done. Conceptually, our results show that a FlexMDM-8B is attainable with approximately (LLaDA-8B pretraining compute) + (modest fine-tuning compute) This suggests that FlexMDM is scalable and could be trained from scratch given sufficient compute!
>
> * Nie et al, Large Language Diffusion Models, 2025.
>
> 4. **Maze task**: As the reviewer noted, our experiments focus solely on the synthetic planning task. We agree that, on their own, these results are insufficient to claim that FlexMDM is categorically superior to MDM on planning tasks. Our intention, however, is to use the maze task primarily as a conceptual testbed; a proof-of-concept demonstration of how flexible-length generation can benefit planning-style reasoning. Consider a reasoning problem that requires multiple intermediate steps, including problem decomposition, verification, and so on. In our synthetic task, each subgoal corresponds to such an intermediate step. FlexMDM handles these naturally because its variable-length generation allows it to insert intermediate steps as needed. In contrast, MDM’s fixed-length structure makes it difficult to learn where to place these steps, causing it to struggle with multi-stage planning.
>
>
> **If our answer addresses/clarifies the reviewer’s concerns, we’re happy to further discuss it!!**

---

> > ### Comment · Reviewer_TnF2 · 2025-11-25
> >
> > Thanks for the rebuttal. My concerns are solved and I will raise my scores accordingly.

---

### Official Review · Reviewer_77ke · 2025-10-30

**Soundness:** 2
**Presentation:** 4
**Contribution:** 3
**Rating:** 6
**Confidence:** 4

**Summary:**

Current masked diffusion models are constrained to fixed-length sample generation, often resorting to tricks (e.g. padding) to deal with variable-length inputs. This paper addresses this issue by introducing a flexible-length masked diffusion model based on token deletion/insertion in addition to masking/unmasking operations. This behavior is achieved by learning to predict an “insertion expectation” (expected number of tokens yet to be inserted after a given position) in addition to the canonical unmasking posterior. The insertion expectation is parameterized directly as an additional scalar prediction head.

The resulting method is able to accurately match the ground-truth sequence length distribution, more so than the fixed-length baseline utilizing padding tokens. It also outperforms the fixed-length baseline on a subgoal-conditioned maze solving task, and can be cheaply retrofitted onto a 8B masked diffusion model (MDM), improving math and coding performance for longer sampling horizons.

**Strengths:**

### S1. Solves an important problem of MDMs
Variable-length generation has been a major limitation of MDMs, and the proposed solution is a valuable addition to both theory and practice of MDMs.

### S2. Theoretically grounded approach
The proposed approach is grounded in theory and provides some guarantees on the likelihood of the trained model.

### S3. Applicability to existing MDMs
The proposed approach can be retrofitted to existing pre-trained MDMs with minimal training (as little as 13B tokens), enabling flexible-length generation for both improved efficiency and accuracy.

**Weaknesses:**

(ordered by descending severity)

### W1. Meager experimental results
While the set of considered tasks is well-rounded, the selective reporting and omission of results as well as non-standard choices that skew the picture in favor of the proposed method need to be rectified:
1. Generative PPL should always be reported together with sequence entropy to control for diversity collapse. Additionally, filtering out short sequences (L1575) is non-standard and skews the picture in favor of FlexMDM. This is an inherent limitation of using the gen. PPL metric and should be controlled for via sequence entropy.
2. Validation loss, and perhaps also training curves, should be reported even if the losses aren’t directly comparable (L416). If the FlexMDM objective indeed is a likelihood bound, then both losses are meaningful and, to some extent, comparable.
3. Accuracy on text benchmarks should also be reported for the sake of consistency and comparison with the literature. These numbers are comparable even if the losses aren’t.
4. Numbers on training/inference speed of FlexMDM vs. MDM are missing. In principle, this should be favorable for FlexMDM due to its smaller average sequence length.
5. The required training time to convert an existing MDM to FlexMDM (L450) should (also) be reported in terms of training tokens, as the wallclock time highly depends on the level of optimization of the given codebase.
6. The reported performance of LLaDA on GSM8k (Fig. 5) is considerably lower than what’s reported in the original paper. The original paper (Nie et al., 2025) reports 70.3% for the base model and 69.4% for the IFT model. It is unclear how this number could get worse by directly training on GSM8k.

### W2. Reproducibility
For the sake of reproducibility, I strongly urge the authors to release all artifacts required to reproduce the results presented in this paper, including trained model checkpoints as well as training and inference code.

### W3. Some fundamental limitations of MDMs remain
While the proposed method enables flexible insertion of tokens during the generation process, some other fundamental limitations remain: Namely, the resulting model is unable to delete or revise existing tokens once filled in. Therefore, the model still risks accumulating errors throughout sampling, just like traditional MDMs.

### Conclusion
Despite the limited experimental results (W1) and concerns regarding reproducibility (W2), the theoretical contributions are strong on their own (S1, S2). Therefore, reasons to accept currently outweigh reasons to reject. The submission can be made significantly stronger still by addressing W1 and W2, and I will be happy to update my final score accordingly.

**Questions:**

- Q1. What are the generative PPLs of MDM and FlexMDM (175M) without filtering by sequence length? (also see W1.1)
- Q2. What is the sequence entropy (as in Zheng et al., 2024) of MDM and FlexMDM (175M) for different numbers of denoising steps? (also see W1.1)
- Q3. What is the training/validation loss (throughout training/final) of MDM and FlexMDM (175M)? (also see W1.2)
- Q4. What is the performance of MDM and FlexMDM (175M) on some relevant text benchmarks (e.g. HellaSwag, ARC-E, ARC-C, WinoGrande, PIQA, etc.)? (also see W1.3)
- Q5. What is the training and inference speed of MDM vs. FlexMDM (both 175M and 8B)? (also see W1.4)
- Q6. How many tokens were used for training LLaDA on FlexMDM? (also see W1.5)
- Q7. Why are the reported numbers on GSM8k (Fig. 5) much lower compared to LLaDA-base (Nie et al., 2025) despite explicitly training on the task? (also see W1.6)
- Q8. Will the model weights and/or training code be released? (also see W2)

Additional questions:
- Q9. Does the FlexMDM training objective constitute a likelihood bound (i.e. ELBO)? If so, this does not appear to be immediately obvious based on L278.
- Q10. Were any alternative ways to parameterize the insertion expectation considered? For example, one may consider modeling the number of insertions as a categorical distribution over the number of to-be-inserted tokens and calculating the expectation thereof.
- Q11. For large-scale pretraining, it is a requirement to have statically-shaped batches for the sake of model compilation and good FLOP utilization. Therefore, in order to apply FlexMDM at scale it will be crucial to have a recipe to do fixed-length training, e.g. through sequence packing. Can FlexMDM support such a scenario?
- Q12. Is it possible to skip the distinct masked case and directly go from empty to unmasked? This could avoid the apparent overhead of first inserting masked tokens and only unmasking them later.
- Q13. Why is a poisson distribution the correct choice for inserting mask tokens? As far as I can tell, this is not explained in the paper.

Nits (not considered for final score):
- Some equations are unnumbered (e.g. L184, L230, L259, L278). For the sake of referenceability, it is good practice to number all equations (and not just the ones that are referenced in the text itself).
- Time notation breaks the convention of t=0 being noise-free and t=1 being complete noise. This is confusing for people familiar with the diffusion literature, especially for those unfamiliar with flow-matching. For the sake of consistency with the literature, I recommend keeping the diffusion convention. Alternatively, a note on this break of convention may help prevent unnecessary confusion.
- L294: Missing parentheses around equation number. Alternatively, it is good practice to reference equations as such, i.e. “Eq. 5” or “Equation 5” or “Eq. (5)”.
- L317: Seemingly missing some words in “Since unmasking indices in an adaptive [inference setting] no longer trace [...]”
- L375: The footnote mark makes the noise schedule appear to be quadratic. Swapping the position of the period and footnote may help, i.e. `$\alpha_t = \beta_t = t$.\footnote{...}`
- L452: Missing period at the end of the line.

---

### References
- Nie et al. (2025): https://arxiv.org/abs/2502.09992
- Zheng et al. (2024): https://arxiv.org/abs/2409.02908

---

> ### Author Response · Authors · 2025-11-24
>
> We appreciate the positive evaluation and pointing out the missing details of the experiment! They are definitely helpful for further improvement of the manuscript’s clarity. Regarding the codebase, we’ll make sure to release **every** pretraining/sampling code and checkpoints (170M MDM, 170M FlexMDM, 8B FlexMDM fine-tuned from LLaDA-8B), for reproducibility.
>
> Regarding the experimental details that the reviewer pointed out, let us address those question-by-question, organizing them into three sections: [Additional evaluations] [Training details] [Theoretical questions].
>
> **[Additional evaluations]**
>
> 1. **Generative PPLs without filtering**: We observe that pretrained MDM generate samples for unusually short sentences, e.g., ["RAW Paste Data"], ["\uff01na\n\nAdvertisements"]. ["Email: steven.aronmangash@yahoo.com"], whereas the pretrained FlexMDM does not generate these short sequences–the minimum length of generated sequences was typically around 10. These short sequences exhibit unusually large Generative PPLs, since it’s calculated as the exponential of the log likelihoods, so it’s easy to blow up in the sequence length < 10 regime. For example, the filtered Generative PPLs of 1024 samples are:
> [272.617, 247.467, 71.95170, 59.489, 48.690, 43.867,...]. (i-th entry indicates the average Generative PPLs across sequences with length longer than i). This is because MDM controls the length implicitly by [PAD] token, failing to capture the length distribution (Figure 4b).
>
> We therefore filtered sequences shorter than 10 for fair comparison of Generative PPLs. Filtering here hence is *not* favorable for FlexMDM, but rather was for the fair comparison.
>
> 2. **Sequence Entropy**: We measure the sequence entropy as proposed in [Zheng et al., 2024] for both models, across different sampling steps. As we can see below, we cannot observe any substantial difference between the entropies of sequences generated from two models, further justifying the fairness of Generative Perplexity evaluation.
>
>
> | Method   | 256     | 512     | 1024    | 2048    | 4096    |
> |----------|---------|---------|---------|---------|---------|
> | MDM      | 4.4732  | 4.5691  | 4.4417  | 4.2259  | 4.1876  |
> | FlexMDM  | 4.4586  | 4.5173  | 4.5660  | 4.4712  | 4.4304  |
>
> *Zheng et al, Masked Diffusion Models are Secretly Time-Agnostic Masked Models and Exploit Inaccurate Categorical Sampling, 2024.
>
> 3. **Training loss / Validation loss**: We also logged our loss functions during training through WandB, so we had access to them. To recall, MDM loss is the unmasking posterior itself, and the FlexMDM loss comprises the unmasking loss/insertion loss. We attach the (training/validation) loss graph vs. training iterations (the global batch size was 1024 across two runs) for two models. The loss scale (\approx 2.0) sanity checks our training since it roughly matches that of [Sahoo et al, 2024].
>
> MDM loss: https://ibb.co/Kxz766Wv
> FlexMDM loss: (training) https://ibb.co/HTWYn3nZ (validation) https://ibb.co/2Yp2gfx1
>
> 4. **Performance on MDM/FlexMDM on text benchmarks**: We evaluate all the standard text benchmarks on our pretrained model: HellaSwag, ARC, MMLU, PIQA. However, we couldn’t really observe performance better than random guessing. (Note that these benchmarks are all the character-level choices; if there are four examples, for instance, the random guessing would give you 25%) We attribute these to two reasons: (1) Models are solely trained on OpenWebText, which mostly consists of news articles, therefore lacking the domain knowledge for downstream benchmarks. (2) Models have never been trained on a prompt-answer type of sequence. Indeed, we couldn’t find any reference to evaluating their OpenWebText pretrained Discrete Diffusion on these text benchmarks, including [Sahoo et al 2024, Shi et al 2024, Arriola et al 2024]. Despite this, we would be happy to include the implementation code of this in our further code release.
>
> *Shi et al. "Simplified and generalized masked diffusion for discrete data, 2024
>
> *Sahoo et al. "Simple and effective masked diffusion language models, 2024
>
> *Arriola et al. "Block diffusion: Interpolating between autoregressive and diffusion language models, 2024

---

> > ### Author Response · Authors · 2025-11-24
> >
> > **[Training details]**
> >
> > 5. **Handling variable-length sequences**: Our dataset is indeed variable length, but we add [PAD] tokens to handle batch arguments for different length sequences, e.g, [  [ a,b,c,[PAD],[PAD] ] , [d,e,f,g,[PAD] ] ]. Whenever we insert/delete tokens, we adjust the number of [PAD] tokens to ensure they remain a B X (max_length) tensor.
> >
> > Given that FlexMDM is designed to receive only clean or masked tokens as inputs, QKV attention should not attend to pad tokens. Current FlashAttention (Dao et al., 2022), however, does not support this for non-causal attention, which is our setup of interest. We therefore implement FlexAttention (Dong et al., 2024) to make it work.
> >
> > *Dao et al. "Flashattention: Fast and memory-efficient exact attention with io-awareness, 2022
> >
> > *Dong et al. "Flex attention: A programming model for generating optimized attention kernels, 2024
> >
> > 6. **Training/Inference speed of MDM/FlexMDM**: As the reviewer pinpointed, a side benefit of using FlexAttention is improved training speed, since FlexMDM performs attention over fewer tokens than MDM’s full-sequence attention (we indeed briefly mention this in lines 1554-1561, as we can see below). For inference, as FlexMDM’s sampling loop requires additional operation (but still with the same number of forward passes), its inference speed is slightly slower than MDM’s.
> > (under the 1024 sampling steps / tau-leaping sampling)
> >
> > The specific training-time and inference-time wall-clock is as follows:
> > | Model   | Training (tokens/sec) | Inference (tokens/sec) |
> > |---------|------------------------|-------------------------|
> > | MDM     | 3.73M                 | 301               |
> > | FlexMDM | 4.70M                 | 273                     |
> >
> > *details: the training speeds are measured under 16H100 NVIDIA GPUs / inference speeds are measured under 1 H100 NVIDIA GPU.
> >
> > 7. **The number of tokens for LLaDA fine-tuning**: As we noted in Appendix F.3, the fine-tuning from LLaDA 8B to FlexMDM 8B is done with (0.2M gradient steps/batch size 64/seqlen 1024). This is an estimate of 13.1B tokens, which is, by contrast, much smaller than the pre-training tokens of LLaDA, 2.3T tokens [Nie et al, 2025]. This demonstrates that the transfer from MDM to LLaDA can easily be done even in an 8B scale.
> >
> > *Nie et al, Large Language Diffusion Models, 2025.
> >
> >
> > 8. **Performance difference on GSM8K**: As we discussed in lines 1692-1694, our evaluation protocol is different from two seminal works (Nie et al, 2025) and (Zhao et al, 2025). Most importantly, (Nie et al (2025) employed four-shot evaluations, whereas our protocol was zero-shot, thus naturally achieving smaller answer accuracy than them. Moreover, for the fair comparison, since our FlexMDM base model hasn’t been supervised fine-tuned, our comparison was (LLaDA 8B-Base + SFT-ed on GSM8K) vs.  (FlexMDM Base + SFT-ed on GSM8K).
> >
> >
> > | Steps | Ours                        | Zhao et al                     | Nie et al      |
> > |-------|------------------------------|---------------------------------|-----------------|
> > | Evaluation model   | LLaDA 8B + SFT-ed on GSM8K | LLaDA 8B-Base + SFT-ed on s1 | LLaDA 8B-Base  |
> > | Evaluation protocol | zero-shot                  | zero-shot                     | four-shots     |
> >
> > *Nie et al, Large Language Diffusion Models, 2025.
> >
> > *Zhao et al, d1: Scaling reasoning in diffusion large language models via reinforcement learning”

---

> > > ### Author Response · Authors · 2025-11-24
> > >
> > > **[Theoretical questions]**
> > >
> > > 9. **ELBO Bound**: We are grateful for the reviewer’s comment. Indeed, minimising the loss function is equivalent to maximising an ELBO. Given the number of equations, we provide the proof in the following link: https://ibb.co/HDGq1Dyw
> > >
> > >
> > > 10. **Alternative parameterization of the insertion expectation**:  As long as we have a model that outputs the insertion expectation, our theory still guarantees that it’s sufficient to sample from the ground truth distribution. Thus, an alternative strategy, e.g., model the categorical distribution over the insertion and then calculate its expectation, is possible. We, however, observe that our scalar head approach with the Bregman divergence loss is more efficient in the sense that it converges faster, and then the actual quantity that we need is the expectation (so that we can just model a one-dimensional scalar for each position).
> > >
> > > 11. **Alternative design of the interpolant design**: Alternative design of the interpolant, e.g, inserting and deleting clean tokens, is also doable (recent work EditFlow [Havasi et al] worked on a similar thing), and these designs subsume in the class of ‘joint interpolant’. However, our specific design of the interpolant has two significant advantages over others: (1) Learning the insertion expectation: Imagine you have a process where we insert clean tokens (not mask tokens). Then we need to learn a more complicated quantity--the distribution over clean tokens. Whereas our design simplifies the learning problem, just learning the `expected’ number of insertions, and using that to just insert mask tokens. (2) Any-order unmasking guarantee: Thanks to our design, we also have an any-order unmasking guarantee (in Section 4)—it’s due to our specific design of the interpolant, and this would not be possible if we used another design of the interpolant. Given that having any-order unmasking can significantly change the downstream performance a lot, this also remains our own advantage compared to other potential modeling strategies.
> > >
> > > *Havasi et al. "Edit Flows: Flow Matching with Edit Operations, 2025.
> > >
> > >
> > > 12. **Poisson distribution**: The main intuition of using the Poisson distribution comes from the idea of tau-leaping discretization. Tau-leaping `batches’ all the events occurring within a fixed interval [t, t+\tau]. Given our rate matrix entries with respect to insertion $\frac{\dot{\alpha}_t}{1 - \alpha_t}$ times $g$, the number of such birth events is therefore the number of arrivals of a homogeneous birth process with this rate.   Therefore, under the tau-leaping approximation, these individual events are replaced by a single draw  $N \sim \mathrm{Poisson}(\lambda_t \, \tau)$.

---

> ### Author Response · Authors · 2025-11-24
>
> Lastly, as mentioned at the beginning of this response, we’ll ensure the release of the entire codebase and all checkpoints. We'll upload those with the official comment to AC. **If you believe that our answer adequately addresses and clarifies your concerns, we are happy to discuss further and kindly ask that you consider raising your score!**

---

> ### Comment · Reviewer_77ke · 2025-11-26
> **Response to the authors' initial rebuttal**
>
> Thank you to the authors for the detailed response and providing additional results to validate the proposed method. Most of my concerns have been addressed, with only one major concern remaining.
>
> 1. It seems that I misunderstood the effect of the filtering, as I was under the impression that short sequences give lower (better) gen. PPL numbers. In light of this, both comparisons are fair, favoring FlexMDM in low-inference budget settings.
> 2. The sequence entropy is a good addition and favors the proposed FlexMDM. While the baseline does not experience catastrophic entropy collapse, the reduction for large numbers of denoising steps is noticeable.
> 3. The provided loss curves are insightful and will be a valuable addition to the paper. However, I am surprised by the big difference in loss magnitude between MDM (converges to ~2) and FlexMDM (converges to ~3). What is the reason for this discrepancy? Do both losses equally bound the likelihood of the data?
> 4. The fact that the trained models perform no better than random guessing on text benchmarks is surprising. For example, this paper (https://arxiv.org/abs/2503.04482) reports better-than-random accuracies on some benchmarks including ARC-E, HellaSwag, and PIQA for MDM in a very similar setting (same model size, also trained on OWT in a purely unconditional setting). Since you seem to have run these benchmarks, could you share the numbers that you obtained? What could be the reason for this discrepancy?
> 5. Makes sense.
> 6. The difference in inference speed points to an unoptimized implementation of the sampling algorithm, which is completely fine for a paper like this and not a fundamental limitation of the proposed method. Nevertheless, it is a limitation that is important to point out in the paper for the sake of transparency and to motivate further investigation into improving this.
> 7. 13.1B tokens is a reasonable amount for model adaptation, reporting this will be more meaningful than wall-clock training times.
> 8. This makes sense and is fine. Perhaps linking to this section of the appendix from the main text will help prevent future confusions.
> 9. I see. It appears that the proposed training objective ($\mathcal{L}\_\theta$) bounds the likelihood up to a constant ($\mathcal{L}\_\*$), which is unknown in practice. Is this an accurate characterization? In this case, the training loss only serves as a surrogate for the log-likelihood and cannot be used directly to estimate the likelihood of the data. While this is completely fine, if my characterization is accurate, this needs to be pointed out for transparency and clarity.
> 10. Makes sense.
> 11. Thank you for elaborating, this makes sense and is a reasonable design decision. I believe this explanation could be a valuable addition to the paper, shedding additional light on why these design decisions were made.
> 12. This makes sense. I believe this argument is missing from the paper and would help justify and clarify the choice of using a Poisson distribution.
>
> In summary, my remaining concern revolves around remaining uncertainty about the downstream performance of the proposed method (point 4) and, in conjunction to this, whether the proposed training objective is a valid way to estimate likelihood (point 9).
> Based on the authors' response, the training loss should indeed be a valid way to do this, in which case I would expect to see better-than-random performance on at least some of the cited NLP benchmarks, especially given the size and training duration of the model. The decision to withhold benchmark numbers despite seemingly having them does not exactly instill confidence in the model's downstream performance either. I would like to stress that even if the numbers turn out to be disadvantageous for FlexMDM, this does not detract from the value and relevance of the paper's contribution and instead serves as important context around the strengths and limitations of the proposed method.
> (It is worth noting at this point that the subgoal-conditioned maze solving task, while creative and valuable, is an incomplete characterization of the model's downstream performance as it is a somewhat contrived scenario that plays into the strengths/weaknesses of FlexMDM vs. MDM.)
>
> In light of this, I am keeping my current score (6; weak accept) until this remaining concern can be resolved.

---

> ### Author Response · Authors · 2025-11-29
>
> We are sincerely grateful to the reviewer for providing a comprehensive direction regarding the argument about the loss. We fully understand the reviewer’s concern, and in response, we provide (1) an additional proof showing a clear ELBO for our loss, and (2) evaluations of pretrained MDM/FlexMDM on downstream text benchmarks.
>
> In summary, for a fixed $x_1$, we provide a proof of $ -\log p_\theta(x_1) \le \text{[FlexMDM loss]}$ if we use Bregman divergence $\phi(x, y) = y - x + x \log \frac{x}{y}$, and we note that that is what is being implemented in the codebase, and we apologize for the confusion. We have updated the manuscript to reflect that. We also provide an evaluation table for pretrained MDM/FlexMDM on downstream text benchmarks, showing non-trivial numbers on certain benchmarks and further explaining (1) why our earlier trials produced less-than-random numbers, and (2) the comparison with https://arxiv.org/abs/2503.04482.
>
> **Additional proof on the ELBO bound** The details of the derivation can be found here https://ibb.co/q34xcddW, which follows by more carefully expanding the terms in the Radon-Nikodym derivative.
>
> **Additional experiment on downstream benchmark** By using this ELBO bound as a proxy of how likely the sentences are, we evaluate two models on the downstream text benchmarks, getting the following results:
>
>
> | Model   | HellaSwag | PIQA  | ARC-E | ARC-C | BoolQ | OBQA | WinoGrade |
> |---------|-----------|-------|-------|-------|-------|------|-----------|
> | FlexMDM | 28.75     | 51.69 | 25.26 | 25.75 | 50.50 | 24.60| 51.78     |
> | MDM     | 29.70     | 52.55 | 25.26 | 26.09 | 50.05 | 23.80| 49.01     |
>
> Although marginal, in certain tasks (PIQA, HellaSwag, WindoGrande), both achieve better-than-random accuracy. Altogether with our proof above, this indicates that our loss indeed serves as a valid candidate for the likelihood proxy.
>
> In addition to this, we provide further details:
>
> [1] Our initial experiments indeed resulted in less-than-random numbers: This is because we previously used the per-token likelihood as the evaluation metric. In particular, we took the model’s likelihood at the answer token and then extracted the answer through it. However, we later found that the per-sequence likelihood, as captured by the loss function, is a better notion for evaluating answers. As we can see, this indeed leads to non-trivial numbers for certain tasks.
>
> [2] Difference from https://arxiv.org/abs/2503.04482: Our numbers on MDM are slightly worse than those reported in the prior work. This is because the prior work’s evaluation is based on summing the likelihood only over clean token positions (i.e., not including the loss on [PAD] tokens, thus not exactly using the loss function). Our FlexMDM, however, does not support this setting due to its insertion operation, and therefore, we use the full loss function of both models as the evaluation metric.
> We hope these points address the reviewer’s concerns, and if there are still unclear aspects, we would be happy to discuss them further. Last but not least, we thank the reviewer again for their careful feedback; our rebuttal certainly enhances our paper’s clarity and value!

---

### Official Review · Reviewer_VJ5y · 2025-10-31

**Soundness:** 3
**Presentation:** 3
**Contribution:** 3
**Rating:** 6
**Confidence:** 4

**Summary:**

This paper proposes **Any-Order Flexible-Length Masked Diffusion (FlexMDM)**, a generalization of masked diffusion models (MDMs) that supports variable-length generation through a continuous-time Markov chain formulation. The authors introduce an *insertion process* parameterized by a learned insertion expectation (g_\theta) in addition to the usual denoising posterior (f_\theta). This enables length-adaptive, any-order generation while preserving theoretical guarantees and compatibility with pretrained MDM weights. Experiments demonstrate strong results on text, maze planning, and code/math reasoning tasks, including scaling to an 8B-parameter model.

**Strengths:**

1. **Clear theoretical formulation.** The continuous-time treatment and derivation of the rate matrix and training objective are elegant and rigorous. The paper provides the right amount of mathematical depth while maintaining readability.
2. **Minimal yet effective extension.** Introducing a scalar insertion expectation per gap is an elegant way to achieve variable-length modeling without compromising the core MDM formulation.
3. **Scalability and empirical validation.** The ability to retrofit large pretrained MDMs and obtain consistent gains across domains (text, code, math) is a strong practical signal.
4. **Well-designed sampler.** The adaptive unmasking and τ-leaping integration are both theoretically justified and empirically efficient.

**Weaknesses:**

0. **Missing support for addition operation**.
Compared to editflow, it supports addition while this work doesn't.

1. **Missing discussion of Seed Diffusion.**
   The paper should reference and compare to *Seed Diffusion: A Large-Scale Diffusion Language Model with High-Speed Inference* (arXiv:2508.02193). Both aim to enable flexible-length or insertion-based generation, but take different approaches—Seed Diffusion uses a single-model canvas growth mechanism, whereas FlexMDM introduces a learned insertion intensity. A brief conceptual comparison would clarify relative advantages (FlexMDM’s theoretical rigor vs. Seed Diffusion’s simplicity).

2. **Apparent need for two models.**
   The description suggests two learned functions—one for denoising and one for insertion—which might appear “not elegant.” It would help to emphasize that these are *two heads of the same model* rather than two independent models. The authors could also discuss whether a unified parameterization (e.g., sharing a latent event-rate representation) is feasible, as done in Seed Diffusion’s single-network formulation.

3. **Limited evaluation in real-world settings.**
   While the presented results are promising, additional experiments on **broader math and coding benchmarks** would strengthen the claims.

   * Include *MATH-500/5000* or *AIME24/25* for compositional reasoning.
   * Extend code evaluation to *HumanEval*, *HumanEval+*, or *MBPP*.
   * Report **pass@10** and **pass@k vs. compute-time** to better capture stochastic sampling performance.

4. **Minor clarity issues.**

   * An architecture figure showing the shared backbone and dual heads would avoid the “two-model” confusion.
   * A brief qualitative example illustrating insertion dynamics would improve intuition.

**Questions:**

See weakenesses.

---

> ### Author Response · Authors · 2025-11-24
>
> We appreciate the reviewer’s positive evaluation and constructive comments. Below we provide our responses:
> 1. **Additional operation**: The reviewer mentioned the missing addition operation, referring to the concurrent work EditFlow [Havasi et al, 2025]. To the best of our understanding, our FlexMDM already supports addition (insertion), as this is the core operation required for variable-length generation. If the reviewer was referring specifically to inserting clean tokens (rather than mask tokens), we highlight the following advantages of our modeling approach:
>
>
> *Havasi et al. "Edit Flows: Flow Matching with Edit Operations, 2025.
>
> - Learning the insertion expectation: If we were to insert clean tokens directly, the model would need to learn a much more complex quantity, the full distribution over clean tokens to insert. Our design simplifies this learning problem by requiring only the expected number of insertions, and we use that to insert mask tokens instead.
>
> - Any-order unmasking guarantee: Our design further offers an any-order unmasking guarantee (discussed in Section 4), enabled by the specific structure of our interpolant. This guarantee would not be attainable with alternative interpolant formulations. Since any-order unmasking can significantly impact downstream performance, this remains a strong advantage of our approach compared to other potential modeling strategies.
>
>
> 2. **Comparison to Seed Diffusion / Modeling differences**: We appreciate the reviewer for pointing out the prior work Seed Diffusion [Song et al, 2025]. As we understand it, Seed Diffusion’s variable-length mechanism relies on block-based decoding, which does not support insertion. While it naturally enables KV caching and may therefore be more computationally efficient, it is unclear that Seed Diffusion actually overcomes the fixed-length limitation of Masked Diffusion Models, because block-based decoding (1) does not allow insertion and (2) breaks the any-order sampling guarantee, as it requires decoding block by block.
>
> *Song et al. "Seed diffusion: A large-scale diffusion language model with high-speed inference, 2025
>
> 3. **Additional evaluations**: In response to the reviewer’s request, we conducted additional evaluations with FlexMDM 8B, focusing on Pass@10 performance on GSM8K and HumanEval-Infill. We continue to observe a consistent advantage of FlexMDM over MDM on these tasks and an interesting tendency where FlexMDM’s performance increases as we put more sampling steps. We provide the results in the link below:
>
>
> GSM8K pass@10: https://ibb.co/d43nN8K5 /
> HumanEval-infill pass@10: https://ibb.co/fYnywpD1
>
> 4. **Clarifications**: We thank the reviewer for highlighting areas that were unclear. We will revise the manuscript accordingly in the next version.
>
>
> **If our answer addresses/clarifies the reviewer’s concerns, we’re happy to further discuss it!!**

---

> > ### Comment · Reviewer_VJ5y · 2025-11-26
> >
> > thank you for the reponse! I'll keep my score.

---

### Official Review · Reviewer_J8NK · 2025-11-03

**Soundness:** 3
**Presentation:** 3
**Contribution:** 3
**Rating:** 8
**Confidence:** 3

**Summary:**

This paper proposes Flexible Masked Diffusion Models (FlexMDM), a discrete diffusion framework that adds token insertion to standard masked diffusion models while provably preserving any-order decoding. It introduces a joint interpolant over token values and indices. The proposed model learns both the usual unmasking posterior and a new insertion expectation, and derives a CTMC-based training objective with guarantees. Empirically, FlexMDM matches MDM perplexity while modeling length distributions far more faithfully. Initializing from a pretrained MDM model (LLaDA-8B), fine-tuned FlexMDM exhibits significant performance improvements on GSM8K and HumanEval-infill as the number of sampling steps increases.

**Strengths:**

- This paper proposes a novel and principled approach to addressing variable-length generation, which is a known limitation of existing masked diffusion models.
- The proposed approach is theoretically grounded: a joint interpolant is defined over both token values and the number of insertions, then a neat training objective is derived based on CTMC, as a natural extension to the standard MDM training loss.
- This paper also proves the compatibility of insertion prediction with adaptive inference in MDM
- The empirical evidence is compelling. FlexMDM models the length distribution much more faithfully than MDM without sacrificing perplexity, and demonstrates better scaling performance on GSM8K and HumanEval-infill

**Weaknesses:**

- It would help practitioners more if the paper could provide more descriptions of the implementation details. E.g., after inserting/deleting tokens, will the padding be dynamically adjusted?
- Some additional questions
    - Figure 3 left: I guess $S_{\tau}$ shoud be $[0, 3, 4]$ right?
    - In algorithm 1, positions are unmasked first before insertion prediction, but I suppose the newly unmasked tokens should not be visible to the insertion module $g_{\theta}$ right?
    - Given the flexibility of combining any-order unmasking with insertion prediction, I wonder whether it's possible to simply reuse the unmasking posterior from a pretrained MDM and only separately learn the insertion expectation.

**Questions:**

Please see Weaknesses

---

> ### Author Response · Authors · 2025-11-24
>
> We are grateful to the reviewer for the positive evaluation and comments. Below, we address the reviewer’s questions:
>
> 1. **Implementation details on the insertion/deletion**: Our dataset is indeed variable length, but we add [PAD] tokens to handle batch arguments for different length sequences, e.g, [  [ a,b,c,[PAD],[PAD] ] , [d,e,f,g,[PAD] ] ]. Whenever we insert/delete tokens, we adjust the number of [PAD] tokens to ensure they remain a B X (max_length) tensor.
>
> 2. **Additional clarifications**: For Figure 3, left, it should be [0,3,4]! And for Algorithm 1, yes, the unmasking/insertion steps happen simultaneously, i.e., the newly inserted tokens aren’t visible to the insertion module. (both modules therefore take $X_{t_k}$ as an input). Regarding the question on the flexibility, it’s not perfectly correct, since FlexMDM additionally needs the unmasking posteriors for shorter sequences than (max_length), going beyond MDM’s unmasking posterior of a length (max_length) sequence. However, once we have a pretrained MDM, getting `additional’ knowledge on the unmasking posterior for shorter sequences isn’t that hard, as we demonstrated in the LLaDA experiment (Section 5.2).
>
>
> We’ll make sure to add these clarifications/revisions in our new version of the manuscript! We’re happy to discuss if there are parts that need further clarification!

---

### Author Response · Authors · 2025-11-29

We appreciate the AC and the reviewers’ efforts in evaluating our paper. Below, we summarize our overall rebuttals and highlight what was addressed during the response period.

**Paper’s contributions and strengths.** All reviewers assessed our paper’s contributions and strengths positively. The fixed-length canvas limitation of masked diffusion hinders variable-length generation and insertion, and FlexMDM directly resolves this by learning insertion expectations, enabling fully flexible any-order generation. We provide rigorous theoretical guarantees for both training and inference, supported by empirical results. Finally, our 8B-scale fine-tuning experiment demonstrates that an 8B FlexMDM is attainable in practice.

**Clarifications**. We carefully addressed all reviewer questions, including detailed experimental setups (77ke) and the treatment of insertion batches (J8NK, TnF2). Please refer to our individual responses for specifics.
Below, we outline the reviewers’ main concerns and how we addressed them:


**Concern 1 (TnF2): Scalability of MDM.**
 Reviewer TnF2 asked what the scaling experiment in Section 5.3 implies. We clarified that only ~0.6% of LLaDA-8B’s pretraining compute is required to retrofit MDM into FlexMDM, demonstrating rapid transfer and confirming FlexMDM’s practical scalability.


**Concern 2 (77ke): FlexMDM loss as a reliable ELBO.**
 Reviewer 77ke asked whether the FlexMDM training loss constitutes an ELBO and how its reliability compares to the MDM counterpart. We showed that the FlexMDM loss is indeed an ELBO with a tractable constant gap. Empirically, we evaluated pretrained FlexMDM and MDM models on OpenWebText and downstream benchmarks, demonstrating that FlexMDM’s training loss remains a faithful proxy for likelihood, comparable to MDM.


We also appreciate the reviewers’ additional suggestions and will upload a refined manuscript by the end of the rebuttal period.

---

### Meta-Review · Area_Chair_fRit · 2026-01-06

**Summary:**

The key decision factors were (i) whether FlexMDM provides a theoretically grounded and practical solution to the fixed-length limitation of masked diffusion models, (ii) whether the proposed insertion modeling and any-order generation can be achieved without sacrificing perplexity or degrading generation quality, and (iii) whether the method is empirically validated at meaningful scale and across diverse tasks, including evidence that it can be efficiently retrofitted from existing MDMs. Reviewers generally agreed the approach is conceptually well-motivated and addresses an important bottleneck for masked diffusion, but they focused their scrutiny on theoretical soundness of the proposed ELBO-style objective, the precise mechanics of insertion/unmasking during sampling, and whether the empirical evaluation adequately characterizes downstream behavior and scalability. The rebuttal and discussion substantially clarified these points and added targeted experiments and diagnostics, leading to broad support for acceptance.

**Reviewer Concerns:**

Addressed concerns: The rebuttal clarified the theoretical foundation by providing a more explicit ELBO proof and tightening the relationship between the FlexMDM objective and likelihood. It also clarified key algorithmic details of insertion modeling (including visibility/conditioning of newly inserted or unmasked tokens, batching/padding behavior under variable length, and what can be reused from pretrained MDMs). Empirically, the authors expanded evaluation with pass@10 results, additional text benchmarks, long-run sampling studies (including behavior over many sampling steps), and auxiliary diagnostics such as entropy and training/validation curves where requested. The rebuttal further strengthened the practical story by demonstrating efficient retrofitting from existing MDMs, including the 8B-scale adaptation results, and by clarifying the compute required for the retrofit relative to pretraining.

Remaining concerns: There is no remaining concern.

**Reviewer Scores:**

J8NK: likely 8
VJ5y: likely 6
77ke: likely 6
TnF2: likely 8

---

### Decision · Program_Chairs · 2026-01-26

Accept (Poster)